# GmGM: a fast Gaussian graphical model for multi-modal data

## Abstract

This paper introduces the Gaussian multi-Graphical Model, a model to construct sparse graph representations of matrix- and tensor-variate data. We generalize prior work in this area by simultaneously learning this representation across several tensors that share axes, which is necessary to allow the analysis of multimodal datasets such as those encountered in multi-omics. Our algorithm uses only a single eigendecomposition per axis, achieving an order of magnitude speedup over prior work in the ungeneralized case. This allows the use of our methodology on large multi-modal datasets such as single-cell multi-omics data, which was challenging with previous approaches. We validate our model on synthetic data and five real-world datasets.

## 1 Introduction

A number of modern applications require the estimation of networks (graphs) exploring the dependency structures underlying the data. In this paper, we propose a new approach for estimating conditional dependency graphs. Two datapoints $x, y$ are *conditionally independent* (with respect to a dataset $\mathcal{D}$) if knowing one provides no information about the other that is not already contained in the rest of the dataset: $\mathbb{P}(x|y, \mathcal{D}_{\backslash xy}) = \mathbb{P}(x|\mathcal{D}_{\backslash xy})$. For normally distributed data, conditional dependencies are encoded in the inverse of the covariance matrix (the 'precision' matrix). Two datapoints are conditionally dependent on each other if and only if their corresponding element in the precision matrix is not zero. If our dataset were in the form of a vector $\mathbf{d}$, we could then model it as $\mathbf{d} \sim \mathcal{N}(\mathbf{0}, \mathbf{\Psi}^{-1})$ for precision matrix $\mathbf{\Psi}$. This is a Gaussian Graphical Model (GGM); $\mathbf{\Psi}$ encodes the graph.

However, datasets are often more structured than vectors. For example, single-cell RNA sequencing datasets (scRNA-seq) come in the form of a matrix of gene expression counts whose rows are cells and columns are genes. Video data naturally requires a third-order tensor of pixels to represent it - rows, columns, and frames. Furthermore multi-omics datasets such as those including both scRNA-seq and scATAC-seq may require two or more matrices to be properly represented; one for each modality.

We could assume that each row of our matrix is an i.i.d. sample drawn from our model. However, independence is a strong and often incorrect assumption. If we wanted to make no independence assumptions, we could vectorize the dataset $\mathcal{D}$ and estimate $\mathbf{\Psi}$ in $\text{vec}[\mathcal{D}] \sim \mathcal{N}(\mathbf{0}, \mathbf{\Psi}^{-1})$. However, this produces intractably large $\mathbf{\Psi}$, whose number of elements is quadratic in the product of the lengths of our dataset's axes.

Thankfully, tensors are highly structured, and we are often interested in the dependency structure of each axis individually - i.e. the dependencies between samples or the dependencies between features - rather than the dependencies between the elements of the tensor themselves. To model this, we can represent $\mathbf{\Psi}$ as some deterministic combination of the axis-wise dependencies: $\text{vec}[\mathbf{D}] \sim$

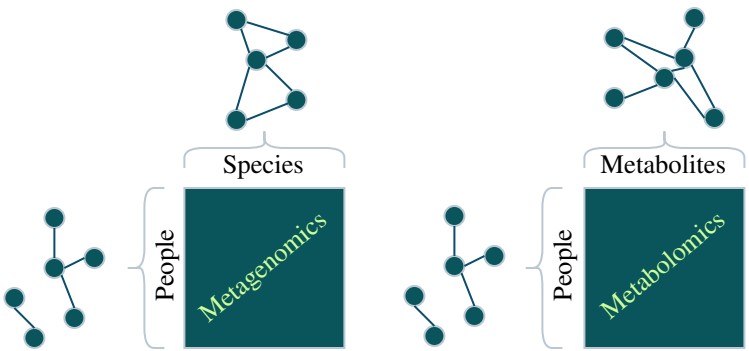

Figure 1: The two matrices of the LifeLines-DEEP dataset. As both matrices include data for the same people, the learned graph between people should be the same.

$\mathcal{N}(\mathbf{0}, \zeta(\boldsymbol{\Psi}_{\mathrm{row}}, \boldsymbol{\Psi}_{\mathrm{col}})^{-1})$, for some function $\zeta$. The strategy is to estimate $\boldsymbol{\Psi}_{\mathrm{row}}, \boldsymbol{\Psi}_{\mathrm{col}}$ directly, without computing the intractable $\zeta(\boldsymbol{\Psi}_{\mathrm{row}}, \boldsymbol{\Psi}_{\mathrm{col}})^{-1}$. While there are multiple choices for $\zeta$, this paper considers only the Kronecker sum.

## 1.1 Prior work

The Kronecker sum BiGraphical Lasso (BiGLasso) model was first considered by Kalaitzis et al. [14]. BiGLasso is the multi-axis analog to graphical lasso methods [10], which are used to estimate covariance matrices of data drawn from a multivariate Gaussian distribution. The Kronecker sum of two matrices, $\mathbf{A} \oplus \mathbf{B}$, can be expressed in terms of Kronecker products: $\mathbf{A} \otimes \mathbf{I} + \mathbf{I} \otimes \mathbf{B}$. When the matrices $\mathbf{A}, \mathbf{B}$ are adjacency matrices of graphs, the Kronecker sum has the interpretation as the Cartesian product of those graphs. This sum is one choice $\zeta$ to combine the per-axis precision matrices into the precision matrix of the vectorized dataset, $\mathrm{vec}[\mathbf{D}] \sim \mathcal{N}(\mathbf{0}, (\boldsymbol{\Psi}_{\mathrm{row}} \oplus \boldsymbol{\Psi}_{\mathrm{col}})^{-1})$.

Other choices for $\zeta$ have been considered, such as using the Kronecker product [23, 8], or the square of the Kronecker sum [24, 25]. Each method has its strengths; the benefits of a Kronecker sum structure are its interpretability as a graph product, stronger sparsity, and its allowance of inter-task transfer [14].

The original BiGLasso model was very slow to converge to a solution, in large part due to its non-optimal space complexity of $O(n^2 p^2)$. This prohibited its use on large datasets (measuring in a couple hundred samples and/or features). Numerous modifications have been made to the algorithm to improve its speed and achieve an optimal space complexity of $O(n^2 + p^2)$, such as scBiGLasso [17], TeraLasso [12], and EiGLasso [27]. Of these, TeraLasso is notable in that it generalizes to an arbitrary number of axes, i.e. $\zeta(\boldsymbol{\Psi}_1, ..., \boldsymbol{\Psi}_k) = \boldsymbol{\Psi}_1 \oplus ... \oplus \boldsymbol{\Psi}_k$. TeraLasso and EiGLasso, the fastest prior work, both rely on computing an eigendecomposition every iteration.

All of these algorithms and models, including our own, rely on a normality assumption. We are most interested in the case of omics data, in which case a log-transform renders our dataset sufficiently Gaussian-like for our algorithm to achieve good performance. An overview of the use of GGMs in omics data is given by Altenbuchinger et al. [2].

## 1.2 Unmet need

Many datasets, especially those in multi-omics, are representable as a collection of matrices or tensors. As a case study, we consider (a subset of) the Lifelines-DEEP dataset from Tigchelaar et al. [22], which is summarized graphically in Figure 1.

In this dataset, two different modalities of data were gathered from the same people: counts of microbial species found in their stools (metagenomics) and counts of metabolites found in their blood plasma (metabolomics). While different matrices, each modality shares an axis. If we were to estimate a graph of people on each modality independently, they would likely yield different graphs. This is not ideal; if our aim is to estimate the true graph of conditional dependencies, there should be only one resultant graph. To estimate it, we should be considering both modalities simultaneously.

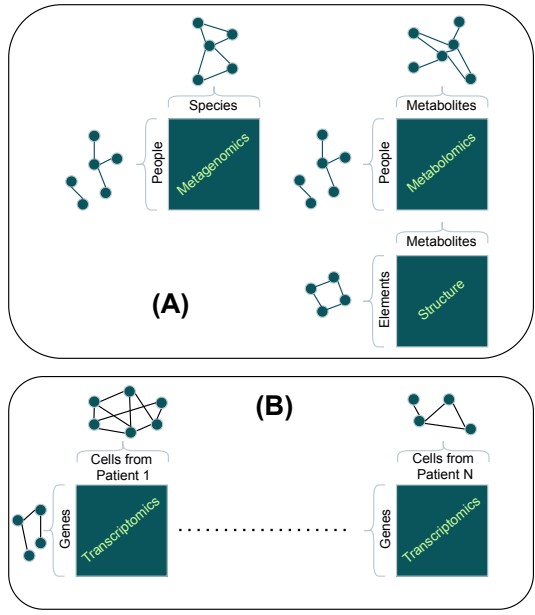

Figure 2: (A) A hypothetical dataset whose structure cannot be reduced to a single tensor by concatenation. Concatenating would lead to a block of missing values for a hypothetical (and nonsensical) species by elements matrix. (B) A hypothetical single-cell RNA-sequencing dataset procured from multiple patients. Concatenation is possible, but would lead to a very large output graph for a modest number of patients.

One way to do this would be to concatenate the modalities, producing a matrix of people by "species+metabolites". This could yield interesting results, if one is interested in connections between individual species and a metabolite. However, it would increase the size of the output graph, which grows quadratically in the length of the axis. Furthermore, it is not always possible or feasible; some datasets may not be concatenatable. We visually demonstrate some cases where concatenation fails in Figure 2.

## 1.3 Our contributions

We introduce a novel method to extend the use of Gaussian Graphical Models to multi-tensor datasets. This extension is essential to model conditional dependencies in multimodal datasets such as those frequently occurring in multi-omics. We present an efficient algorithm to estimate these conditional dependencies. When restricted to the single-tensor case, our algorithm is much faster than previous algorithms that estimated conditional dependency graphs for each axis, such as TeraLasso[12] and EiGLasso[27].

# 2 Methods

## 2.1 Notation

In prior work, a single-tensor dataset $\mathcal{D}$ is modelled as $\text{vec}\,[\mathcal{D}] \sim \mathcal{N}\left(\mathbf{0}, \left(\bigoplus_\ell \mathbf{\Psi}_\ell\right)^{-1}\right)$, also written as $\mathcal{D} \sim \mathcal{N}_{KS}\left(\{\mathbf{\Psi}_\ell\}_\ell\right)$.

Our model considers multiple tensors, each with their own (potentially shared) axes. We aim to estimate the precision matrices $\mathbf{\Psi}_\ell$ for each axis $\ell$ of each tensor $\mathcal{D}^\gamma$, indexed by $\gamma \in \mathbb{N}$. To describe that an axis $\ell$ is one of the axes of a tensor $\mathcal{D}^\gamma$, we will write $\ell \in \gamma$. Some values will be indexed by both an axis and a tensor; for consistency we will use subscripts to denote axes (typically $\ell$) and superscripts to denote tensors (typically $\gamma$). $d_\forall^\gamma$ will represent the number of elements in $\mathcal{D}^\gamma$, and $d_\forall = \sum_\gamma d_\forall^\gamma$.

96 An important concept is the Gram matrix $\mathbf{S}_\ell^\gamma$. In the single-tensor case, this is a sufficient statistic; all
97 prior work first computes these matrices as the first step in their algorithm. Let $\mathrm{mat}_\ell\left[\mathcal{D}^\gamma\right]$ represent
98 the "matricization" of $\mathcal{D}^\gamma$ along axis $\ell$, then $\mathbf{S}_\ell^\gamma = \mathrm{mat}_\ell\left[\mathcal{D}^\gamma\right]\mathrm{mat}_\ell\left[\mathcal{D}^\gamma\right]^T$. The matricization of a
99 tensor picks one axis, $\ell$, to index the rows, and flattens the rest out into columns. Note that for
100 a matrix $\mathbf{M}$, $\mathrm{mat}_{\mathrm{columns}}\left[\mathbf{M}\right] = \mathbf{M}^T$. Rather than $\mathbf{S}_\ell^\gamma$, we consider the "effective Gram matrices"
101 $\mathbf{S}_\ell = \sum_{\gamma|\ell\in\gamma}\mathbf{S}_\ell^\gamma$, as these fulfill the role of the Gram matrices in the multi-tensor case.

## 2.2 The model

103 To properly handle sets of tensors, we propose modelling each tensor as being drawn independently
104 from a Kronecker-sum normal distribution. If the tensors share an axis $\ell$, then they will still be drawn
105 independently - but their distributions will be parameterized by the same $\mathbf{\Psi}_\ell$. For an arbitrary set of
106 tensors, the model is:

$$\mathcal{D}^\gamma \sim \mathcal{N}_{KS}\left(\{\mathbf{\Psi}_\ell\}_{\ell\in\gamma}\right)$$
$$\text{for } \mathcal{D}^\gamma \in \{\mathcal{D}^\gamma\}_\gamma$$

107 We call this model the "Gaussian multi-Graphical Model" (GmGM) as it extends Gaussian Graphical
108 Models to estimate multiple graphs from a set of tensors. In this paper, we will make the assumption
109 that no tensor in our set contains the same axis twice - notably, covariance matrices would violate
110 this assumption. Any tensor with a repeated axis would naturally be interpretable as a graph - such
111 datasets are rare, and if one already has a graph the need for an algorithm such as this is diminished.

112 As an example, we model the LifeLines-DEEP dataset $\mathbf{D}^{\mathrm{metagenomics}}$ and $\mathbf{D}^{\mathrm{metabolomics}}$ indepen-
113 dently as:

$$\mathbf{D}^{\mathrm{metagenomics}} \sim \mathcal{N}_{KS}\left(\mathbf{\Psi}_{\mathrm{people}}, \mathbf{\Psi}_{\mathrm{species}}\right)$$
$$\mathbf{D}^{\mathrm{metabolomics}} \sim \mathcal{N}_{KS}\left(\mathbf{\Psi}_{\mathrm{people}}, \mathbf{\Psi}_{\mathrm{metabolites}}\right)$$

## 2.3 The algorithm

115 Here, we present an algorithm to compute the maximum likelihood estimate (MLE) jointly for all
116 parameters $\mathbf{\Psi}_\ell$ of the GmGM. The general idea is to produce an analytic estimate for the eigenvectors
117 of $\mathbf{\Psi}_\ell$, and then iterate to solve for the eigenvalues; this is summed up graphically in Figure 3.

118 In the supplementary material, we derive the following:

$$p(\{\mathcal{D}^\gamma\}) = \frac{\prod_\gamma \sqrt{\left|\bigoplus_{\ell\in\gamma}\mathbf{\Psi}_\ell\right|}}{(2\pi)^{\frac{d_\forall}{2}}} e^{\frac{-1}{2}\sum_\ell \mathrm{tr}[\mathbf{\Psi}_\ell\mathbf{S}_\ell]} \qquad \text{(pdf of GmGM)}$$

$$\mathrm{NLL}\left[\{\mathcal{D}^\gamma\}\right] \propto \sum_\ell \mathrm{tr}\left[\mathbf{\Psi}_\ell\mathbf{S}_\ell\right] - \sum_\gamma \log\left|\bigoplus_{\ell\in\gamma}\mathbf{\Psi}_\ell\right| \qquad \text{(negative log likelihood)}$$

119 From this, we can observe that the effective Gram matrices $\mathbf{S}_\ell$ form a set of sufficient statistics for
120 our distribution. Furthermore, the log-likelihood is the sum of log-likelihoods in the single-axis case,
121 thus preserving convexity of the loss function.

122 **Theorem 1.** *Let $\mathbf{V}_\ell\mathbf{e}_\ell\mathbf{V}_\ell^T$ be the eigendecomposition of $\mathbf{S}_\ell$ (where $\mathbf{V}_\ell \in \mathbb{R}^{d_\ell\times d_\ell}$ and $\mathbf{e}_\ell \in \mathbb{R}^{d_\ell\times d_\ell}$*
123 *is a diagonal matrix). Then $\mathbf{V}_\ell$ are the eigenvectors of the maximum likelihood estimate of $\mathbf{\Psi}_\ell$.*

124 Theorem 1 is critical to allowing efficient estimation of $\mathbf{\Psi}_\ell$, as it not only allows us to extract the
125 computationally intensive eigendecomposition operation from the iterative portion of the algorithm,
126 but also reduces the number of parameters to be linear in the length of an axis.

127 To find the eigenvalues $\mathbf{\Lambda}_\ell$ of $\mathbf{\Psi}_\ell$, we produce the second theorem:

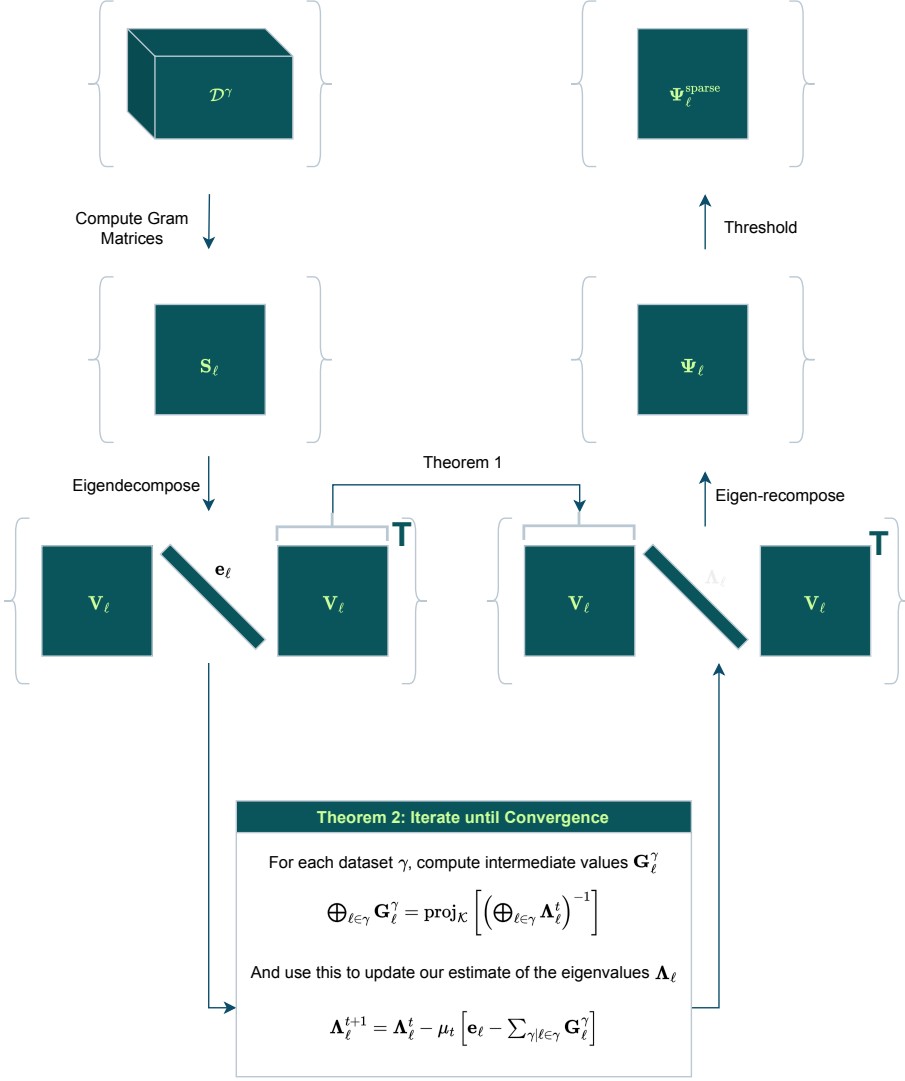

Figure 3: A graphical overview of how the GmGM algorithm works. We use $\gamma$ to represent an arbitrary modality, and $\ell$ to represent an arbitrary axis. Proofs are given in the supplementary material.

**Theorem 2.** *Let $\{\mathbf{G}_\ell^\gamma\}$ be matrices such that the expression $\bigoplus_{\ell \in \gamma} \mathbf{G}_\ell^\gamma$ is the best Frobenius-norm approximation of $\left(\bigoplus_{\ell \in \gamma} \mathbf{\Lambda}_\ell^t\right)^{-1}$. Then, for a learning rate $\mu_t$, gradient descent can be performed with the update equation $\mathbf{\Lambda}_\ell^{t+1} = \mathbf{\Lambda}_\ell^t - \mu_t \left[\mathbf{e}_\ell - \sum_{\gamma | \ell \in \gamma} \mathbf{G}_\ell^\gamma\right]$. As $\mathbf{\Psi}_\ell$ is positive definite, $\mu_t$ must be chosen to prevent $\mathbf{\Lambda}_\ell^t$ from becoming negative.*

While the definition of $\mathbf{G}_\ell^\gamma$ is technical, it is analogous to the notion of the blockwise-trace from Kalaitzis et al. [14] and $\text{proj}_\mathcal{K}$ from Greenewald, Zhou, and Hero III [12]. Proofs of Theorems 1 and 2, along with a method to compute $\mathbf{G}_\ell^\gamma$, are given in the supplementary material. Overall, our algorithm is described in the pseudocode at the top of the next page.

For regularization, one can choose to either keep the top $p\%$ of edges, or keep the top $k$ edges per vertex (for parameters $p, k$). The incorporation of more advanced regularizers, such as Lasso, would require an eigen-recomposition on each iteration, which would be much slower. As we demonstrate empirically in Section 3, it is not necessary to use advanced regularizers to recover the graph structure to the same precision as prior work.

---

**The GmGM algorithm**

---

**Input:** $\{\mathcal{D}_i^\gamma\}$, tolerance
**Output:** $\{\mathbf{\Psi}_\ell\}$
1: **for** $1 \le \ell \le K$
2: $\quad \mathbf{S}_\ell \leftarrow \sum_{\gamma|\ell\in\gamma} \frac{1}{n^\gamma} \sum_i^{n^\gamma} \mathrm{mat}_\ell\left[\mathcal{D}_i^\gamma\right] \mathrm{mat}_\ell\left[\mathcal{D}_i^\gamma\right]^T$
3: $\quad \mathbf{V}_\ell \leftarrow \mathrm{eigenvectors}[\mathbf{S}_\ell]$
4: $\quad \mathbf{e}_\ell \leftarrow \mathrm{eigenvalues}[\mathbf{S}_\ell]$
5: **end for**
6: $\mathbf{\Lambda} \leftarrow \begin{bmatrix} 1 & ... & 1 \end{bmatrix}^T$
7: $\mu \leftarrow 1$
8: **while** not converged
9: $\quad$ **for** $1 \le \ell \le K$
10: $\quad\quad \mathbf{G}_\ell^\gamma \leftarrow \mathrm{proj}_{KS}\left[\left(\bigoplus_{\ell'\in\gamma} \mathbf{\Lambda}_\ell\right)^{-1}\right]$
11: $\quad\quad \mathbf{\Lambda}_\ell' \leftarrow \mathbf{\Lambda}_\ell - \mu\left[\mathbf{e}_\ell - \sum_{\gamma|\ell\in\gamma} \mathbf{G}_\ell^\gamma\right]$
12: $\quad$ **end for**
13: $\quad$ **for** $1 \le \ell \le K$
14: $\quad\quad \mathbf{\Lambda}_\ell \leftarrow \mathbf{\Lambda}_\ell'$
15: $\quad$ **end for**
16: $\quad$ **for** $\gamma$
17: $\quad\quad$ **if** $\sum_{\ell\in\gamma} \min\mathbf{\Lambda}_\ell < \mathrm{tolerance}$ **then**
18: $\quad\quad\quad$ decrease $\mu$ so that this result is sufficiently far from zero
19: $\quad\quad$ **end if**
20: $\quad$ **end for**
21: **end while**
22: **for** $1 \le \ell \le K$
23: $\quad \mathbf{\Psi}_\ell \leftarrow \mathbf{V}_\ell \mathbf{\Lambda}_\ell \mathbf{V}_\ell^T$
24: **end for**

---

## 3 Results

We tested our algorithm on synthetic data and five real-world datasets. Explanations of data generation, collection, preprocessing, and regularization are given in the supplementary material.

### 3.1 Synthetic Data

We verified that our algorithm was indeed faster on matrix-variate data compared to prior work (Figure 4) on our computer (Ubuntu 20.04 with Intel Core i7 Processor and 8GB RAM). Our results on matrix data are encouraging - extrapolating the runtimes, datasets up to size 16,000 by 16,000 could have their graphs estimated in less than an hour. Larger datasets would require more than 6GB of memory for our algorithm to run, pushing the limits of RAM. Our algorithm was not significantly faster on higher-order tensor data (see the supplementary material). This is due to the complexity of computing the Gram matrices, which grows exponentially with the number of axes.

In addition to these speed improvements, we show that we perform equivalently to state-of-the-art on matrix data (Figure 5a). On higher-order tensor data, we are outperformed by TeraLasso, which is able to achieve near-perfect recovery of the graphs. We believe this is due to our algorithm's use of thresholding rather than a more advanced regularization technique. Since our speed gains are not significant relative to TeraLasso, on higher-order tensor data without shared axes one should prefer TeraLasso to GmGM. Finally, we demonstrate that taking into account shared axes does indeed improve performance (see blue line, Figure 5b). Prior work could not take this into account.

### 3.2 Real Data

We tested our method on various real datasets. These include two video datasets (COIL-20 [19] and EchoNet-Dynamic [20]), a transcriptomics dataset (E-MTAB-2805 [5]), and two multi-omics datasets (LifeLines-DEEP [22] and a 10x Genomics dataset [1]).

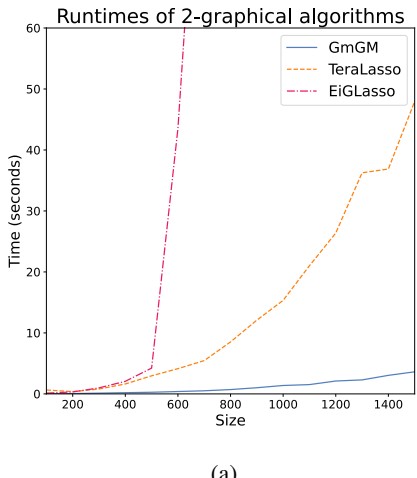

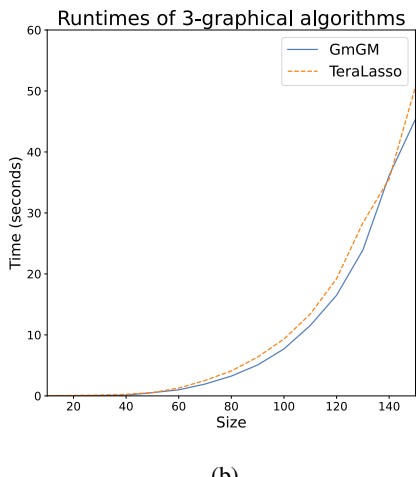

Figure 4: A comparison of the runtimes of our algorithm against (a) bi-graphical and (b) tensor-graphical prior work. Runtimes were averaged over 5 runs.

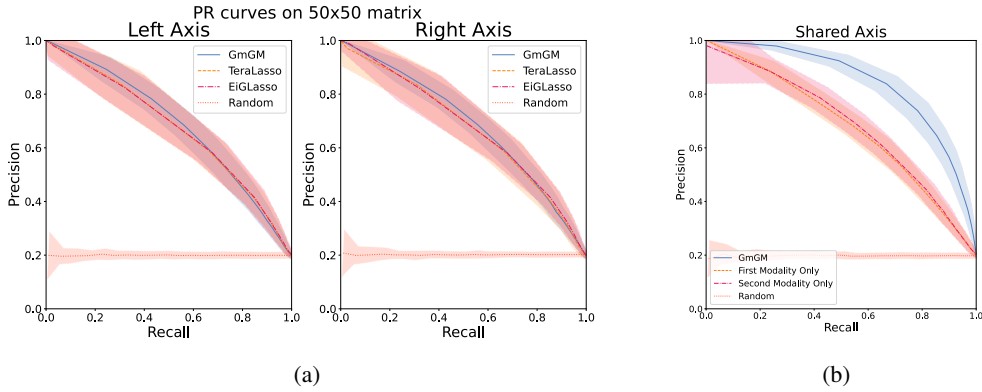

Figure 5: (a) Precision-recall curves comparing various algorithms on synthetic 50x50 matrix data. (b) Precision-recall curves comparing our algorithm on two 50x50 matrices with one shared axis. We considered both modalities simultaneously (blue) and an individual modality (red, orange). In both subfigures, each edge of the true graphs was generated independently with probability $\frac{1}{5}$.

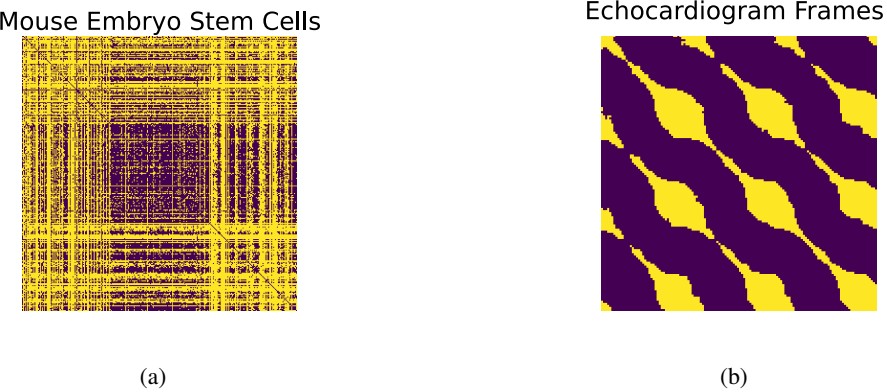

Figure 6: The estimated precision matrices on the E-MTAB-2805 dataset (a) and the EchoNet-Dynamic dataset (b). Yellow represents an edge and purple represents the lack of an edge. The E-MTAB-2805 cells have been grouped together by cell cycle stage, in the order G, S, and G2/M.

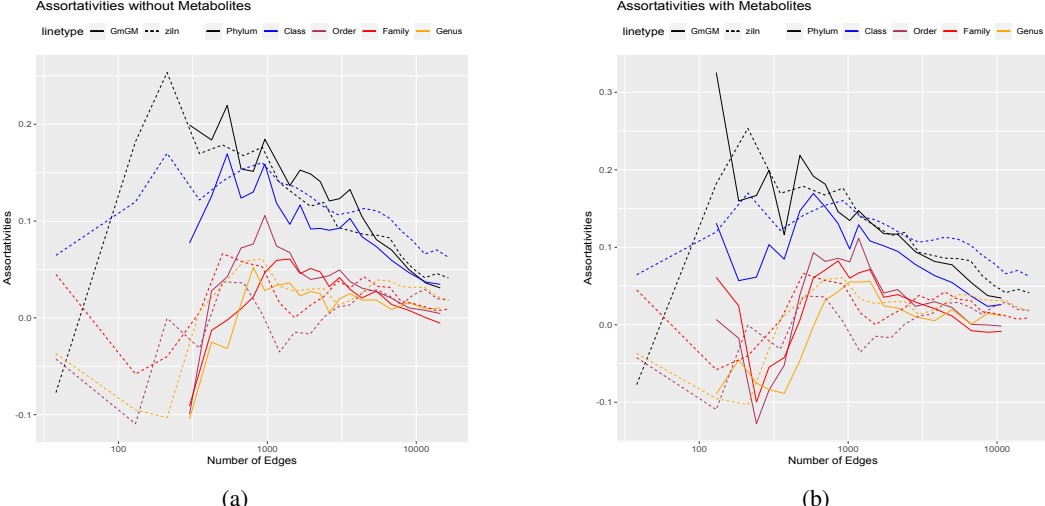

(a)  (b)

Figure 7: Assortativity with increasing regulatization in the LifeLines-DEEP dataset, comparing our method with the Zero-inflated Log-Normal (ZiLN) model. In one case we show the performance of our algorithm restricted to the metagenomics dataset (a) and when augmented with the metabolomics dataset (b). In both cases, ZiLN is only trained on the metagenomics dataset, as it is a single-axis model.

The E-MTAB-2805 dataset consists of transcriptomics data for individual cells split into three groups by their stage in the cell cycle (G, S, and G2/M). If our estimated precision matrices had a 3x3 block-diagonal structure, this would indicate that it had recreated this grouping. This is not what we see, but we do see a 3x3 block matrix structure (Figure 6a). We found that cells in the DNA synthesis stage (S) had few connections between them, and that there were many connections between the G1 and G2/M stages. This result is biologically plausible, as cells in the synthesis stage are the most variable.

The results on EchoNet-Dynamic (Figure 6b) are much more encouraging, as we would expect a periodic structure due to the beating of the heart. A precision matrix with repeating diagonals is what we would expect to see in this case, which is what our algorithm produces. In the supplementary material, we further verify that this corresponds to a heartbeat by using the repetition to accurately predict the opening of the mitral valve in the video.

The duck video in the COIL-20 dataset was considered in the original BiGLasso paper [14], in which they showed that their algorithm could recover the ordering of the frames of the video. To do this they had to heavily downsample the image (to a 9x9 image with half the frames), and flatten the rows and frames into a single axis. Due to the speed improvements of our algorithm, and its ability to handle tensor-variate data, we were able to run our algorithm on the raw, unprocessed data and achieve a similar result in negligible time. Specifically, the reconstruction of the frames had an accuracy of 99%.

Prior work by Prost, Gazut, and Brüls [21] used assortativity to assess their validity of the species graph estimated by their model on the LifeLines-DEEP metagenomics dataset. Assortativity represents the tendency of related species to cluster together in the graph. A random graph would have an assortativity of zero, but we would expect moderate assortativity in the true network as similar species may fulfill similar roles in the gut microbiome. Our assortativity is comparable to prior work (Figure 7). We also found that our graphs were more robust to noise than prior work; we analyze this in the supplementary material.

Finally, we tested our approach on a 10x Genomics single-cell (RNA+ATAC) dataset taken from a B Cell lymphoma tumour. We demonstrate that the clusters we find (using Louvain clustering[3]) on the graph remain visually cohesive when projected into lower-dimensional space by UMAP[18] (Figure 8). In particular, the disconnected "islands" in UMAP correspond to their own cluster on the graph as well. As these island-clusters were arrived at independently through two methods, UMAP and our algorithm, it increases our confidence in the validity of the clustering. In the supplementary

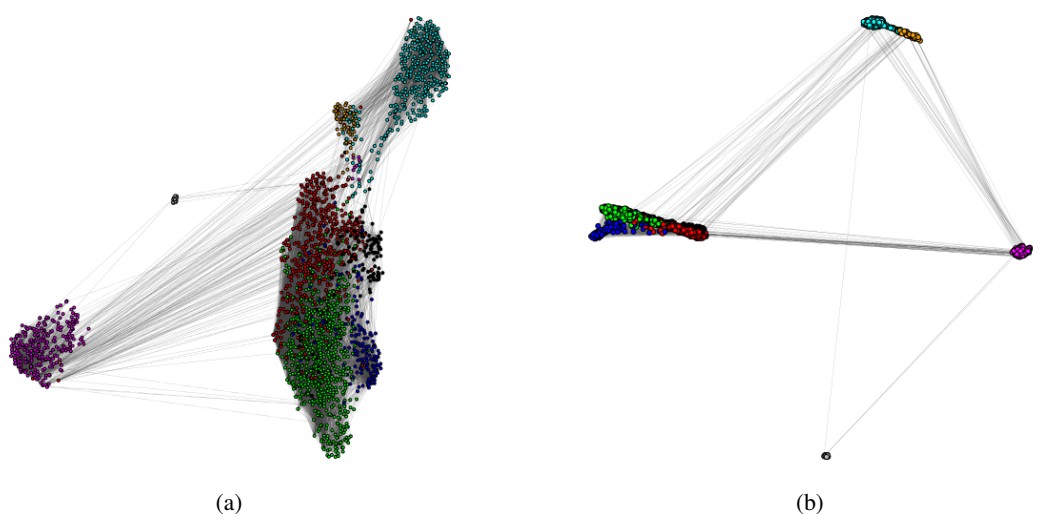

(a)            (b)

Figure 8: Two plots of the same cells from the 10x Genomics dataset, displayed via UMAP [18] (a) and the Fruchterman-Reingold layout algorithm[11] (b). Colors are based on Louvain clustering[3] of the graph, and represent the same clustering in both figures.

material, we verify that these clusters do represent distinct groups via a GO term enrichment analysis. Our overall approach has been implemented in Python. All of the code to run the algorithm and recreate the experiments has been made publicly available on GitHub; https://github.com/NeurIPS-GmGM-Paper/GmGM.

## 4   Limitations

Our method uses thresholding rather than more sophisticated regularizers. However, there is no fundamental barrier preventing our algorithm from allowing regularizers at the cost of an eigen-recomposition per iteration. This would increase the asymptotic complexity of the iterative portion of our algorithm, making it questionable whether any gains in precision would be worth the loss in efficiency.

Our method assumes that no tensor has a repeated axis (i.e. a matrix of people by people rather than people by species). If there is a repeated axis, one can no longer analytically find the eigenvectors of the MLE, at least by our methods. This is not a substantial issue, as such datasets are uncommon and already represent graphs. Rather than extending the algorithm to work with repeated-axis tensors, it would be more fruitful to extend it to work with priors.

When considering multi-tensor datasets, it may be the case that two axes only partially overlap. For example, the full LifeLines-DEEP dataset contains a second (follow-up) metagenomics dataset for a third of the study participants; two thirds of the patients are missing from this dataset. We do not make an attempt to handle this type of missing data, even though missing data shows up in many applications. The lack of ability to handle missing data is a major limitation of our algorithm. It is nontrivial to extend the algorithm to handle this case, as it renders Theorem 1 ineffective and hence removes the speed advantage we attained. Prior work has not addressed this problem, as it only exists in multi-tensor datasets and we are the first to consider this case.

## 5   Conclusion

We have created a novel model, GmGM, which successfully generalizes Gaussian graphical models to the common scenario of multi-tensor datasets. Furthermore, we demonstrated that our algorithm is significantly faster than prior work focusing on Gaussian tensor-graphical models such as EiGLasso and TeraLasso while still preserving state-of-the-art performance. These speed improvements allow tensor-graphical models to be applied to datasets with axes of length in the thousands. Finally, we demonstrated the application of our algorithm on five real-world datasets to prove its efficacy.

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
