# Supplementary material to
# GmGM: a fast Gaussian graphical model for multi-modal data

# Contents

## 1 Notation

In addition to the notation used in the main paper, we also introduce further notation to aid in the proofs. For working with tensors, Kolda and Bader [8] proved to be an invaluable resource; we have borrowed their notation in most cases. The only exception is that we have chosen to denote the $\ell$-mode matricization of a tensor $\mathcal{T}$ as $\mathrm{mat}_\ell\left[\mathcal{T}\right]$ rather than $\mathcal{T}_{(\ell)}$, to highlight its similarity to $\mathrm{vec}\left[\mathcal{T}\right]$ and free up the subscript for other purposes.

Submitted to 37th Conference on Neural Information Processing Systems (NeurIPS 2023). Do not distribute.

24     To keep track of lengths of axes, we define the following notation:

25         • $d_\ell^\gamma$ is the length of axis $\ell$

26         • $d_{>\ell}^\gamma$ is the product of lengths of all axes after $\ell$

27         • $d_{<\ell}^\gamma$ is the product of lengths of all axes before $\ell$

28         • $d_{\backslash\ell}^\gamma$ is the product of lengths of all axes except for $\ell$

29         • $d_\forall^\gamma$ is the product of lengths of all axes (i.e. the number of elements in $\mathcal{D}^\gamma$)

30         • $d_\forall = \sum_\gamma d_\forall^\gamma$ is the total number of elements across all datasets

31     In prior work, $d_\ell$ has been used to represent the lengths of axes but $m_\ell$ was used where we write $d_{\backslash\ell}$
32     (such as in [5]). As prior work also used $\backslash\ell$ to represent leaving out the $\ell$th axis in other contexts
33     (such as in [6]), and the analogous definitions of $d_{>\ell}$ and $d_{<\ell}$ were convenient for use in proofs, we
34     chose to introduce $d_{\backslash\ell}$ as the variable to represent leave-one-out length products. By representing all
35     of these related concepts with similar symbols, we hope the maths will be easier to parse.

36     We will let $\mathbf{I}_a$ be the $a \times a$ identity matrix, which allows a concise definition of the Kronecker sum:
37     $\bigoplus_\ell \boldsymbol{\Psi}_\ell = \sum_\ell \mathbf{I}_{d_{<\ell}} \otimes \boldsymbol{\Psi}_\ell \otimes \mathbf{I}_{d_{>\ell}}$.

38     We make frequent use of the vectorization $\mathrm{vec}\,[\mathbf{M}]$ of a matrix $\mathbf{M}$, and more generally of a tensor
39     $\mathrm{vec}\,[\mathcal{T}]$. We adopt the rows-first convention of vectorization, such that:

$$\mathrm{vec} \begin{bmatrix} 1 & 2 \\ 3 & 4 \end{bmatrix} = \begin{bmatrix} 1 & 2 & 3 & 4 \end{bmatrix} \tag{1}$$

40     While columns-first is more common, rows-first is more natural when we adopt the convention that
41     rows are the first axis of tensor; this is the convention that matricization uses, and matricization
42     is much more important for our work due to its role in defining the Gram matrices. Note that, for
43     matrices, a rows-first vectorization of $\mathbf{M}$ is equivalent to a columns-first vectorization of $\mathbf{M}^T$, so there
44     is no fundamental difference between the two. For vectorizing a tensor, we proceed by stacking the
45     rest of the axes in order, such that an element $(i_1, ..., i_K)$ in $\mathcal{T}$ gets mapped to the element $\sum_\ell i_\ell d_{<\ell}$
46     in $\mathrm{vec}\,[\mathcal{T}]$.

47     We define the Gram matrices as $\mathbf{S}_\ell^\gamma = \mathrm{mat}_\ell\,[\mathcal{D}^\gamma]\,\mathrm{mat}_\ell\,[\mathcal{D}^\gamma]^T$. Typically we consider only the one-
48     sample case but if you have multiple samples, indexed by a subscript $i$, then the Gram matrix becomes
49     an average: $\mathbf{S}_\ell^\gamma = \frac{1}{n} \sum_i^n \mathrm{mat}_\ell\,[\mathcal{D}_i^\gamma]\,\mathrm{mat}_\ell\,[\mathcal{D}_i^\gamma]^T$.

50     An essential concept is that of the "stridewise-blockwise trace", defined as:

$$\mathrm{tr}_b^a\,[\mathbf{M}] = \left[ \mathrm{tr}\,\left[ \mathbf{M}\,\left( \mathbf{I}_a \otimes \mathbf{J}^{ij} \otimes \mathbf{I}_b \right) \right] \right]_{ij} \tag{2}$$

51     Where $\mathbf{J}^{ij}$ is the matrix of zeros except at $(i, j)$ where it has a 1. It is a generalization of the
52     blockwise trace considered by Kalaitzis et al. [6], and is related to the $\mathrm{proj}_\mathcal{K}$ operation defined by
53     Greenewald, Zhou, and Hero III [5]. Specifically, $\mathrm{proj}_\mathcal{K}\,[\mathbf{M}]$ is equivalent to $\bigoplus_\ell \mathrm{tr}_{d_{>\ell}}^{d_{<\ell}}\,[\mathbf{M}]$ up to
54     an additive diagonal factor (Lemma 33 from Greenewald, Zhou, and Hero III [5]). $\mathrm{proj}_\mathcal{K}\,[\mathbf{M}]$ was
55     defined to be the matrix that best approximates $\mathbf{M}$ (in terms of the Frobenius norm) while being
56     Kronecker-sum-decomposable. This matrix is not unique; the choice by Greenewald, Zhou, and
57     Hero III [5] to include an additive factor was to enforce $\mathrm{tr}\,[\mathrm{proj}_\mathcal{K}\,[\mathbf{M}]] = 0$. We do not wish to
58     enforce this constraint as it would be impossible to preserve in the multi-tensor case.

59     The parameter $b$ of the stridewise-blockwise trace partitions the $m \times m$ matrix $\mathbf{M}$ into a block matrix
60     with $b \times b$ blocks of size $(\frac{m}{b} \times \frac{m}{b})$. The parameter $a$ then partitions these blocks into a "strided"
61     matrix with $a \times a$ strides containing $\frac{m}{ab} \times \frac{m}{ab}$ blocks. We take the trace of each stride, and the final
62     matrix is the matrix of these traces. As this is conceptually complicated, we provide an example.

$$\text{tr}_2^2 \begin{bmatrix} 1 & 2 & 3 & 4 & 5 & 6 & 7 & 8 \\ 1 & 2 & 3 & 4 & 5 & 6 & 7 & 8 \\ 1 & 2 & 3 & 4 & 5 & 6 & 7 & 8 \\ 1 & 2 & 3 & 4 & 5 & 6 & 7 & 8 \\ 1 & 2 & 3 & 4 & 5 & 6 & 7 & 8 \\ 1 & 2 & 3 & 4 & 5 & 6 & 7 & 8 \\ 1 & 2 & 3 & 4 & 5 & 6 & 7 & 8 \\ 1 & 2 & 3 & 4 & 5 & 6 & 7 & 8 \end{bmatrix} \tag{3}$$

$$= \text{tr}^2 \begin{bmatrix} \text{tr}\begin{bmatrix} 1 & 2 \\ 1 & 2 \end{bmatrix} & \text{tr}\begin{bmatrix} 3 & 4 \\ 3 & 4 \end{bmatrix} & \text{tr}\begin{bmatrix} 5 & 6 \\ 5 & 6 \end{bmatrix} & \text{tr}\begin{bmatrix} 7 & 8 \\ 7 & 8 \end{bmatrix} \\ \text{tr}\begin{bmatrix} 1 & 2 \\ 1 & 2 \end{bmatrix} & \text{tr}\begin{bmatrix} 3 & 4 \\ 3 & 4 \end{bmatrix} & \text{tr}\begin{bmatrix} 5 & 6 \\ 5 & 6 \end{bmatrix} & \text{tr}\begin{bmatrix} 7 & 8 \\ 7 & 8 \end{bmatrix} \\ \text{tr}\begin{bmatrix} 1 & 2 \\ 1 & 2 \end{bmatrix} & \text{tr}\begin{bmatrix} 3 & 4 \\ 3 & 4 \end{bmatrix} & \text{tr}\begin{bmatrix} 5 & 6 \\ 5 & 6 \end{bmatrix} & \text{tr}\begin{bmatrix} 7 & 8 \\ 7 & 8 \end{bmatrix} \\ \text{tr}\begin{bmatrix} 1 & 2 \\ 1 & 2 \end{bmatrix} & \text{tr}\begin{bmatrix} 3 & 4 \\ 3 & 4 \end{bmatrix} & \text{tr}\begin{bmatrix} 5 & 6 \\ 5 & 6 \end{bmatrix} & \text{tr}\begin{bmatrix} 7 & 8 \\ 7 & 8 \end{bmatrix} \end{bmatrix} \tag{4}$$

$$= \text{tr}^2 \begin{bmatrix} 3 & 7 & 11 & 15 \\ 3 & 7 & 11 & 15 \\ 3 & 7 & 11 & 15 \\ 3 & 7 & 11 & 15 \end{bmatrix} \tag{5}$$

$$= \begin{bmatrix} \text{tr}\begin{bmatrix} 3 & 11 \\ 3 & 11 \end{bmatrix} & \text{tr}\begin{bmatrix} 7 & 15 \\ 7 & 15 \end{bmatrix} \\ \text{tr}\begin{bmatrix} 3 & 11 \\ 3 & 11 \end{bmatrix} & \text{tr}\begin{bmatrix} 7 & 15 \\ 7 & 15 \end{bmatrix} \end{bmatrix} \tag{6}$$

$$= \begin{bmatrix} 14 & 22 \\ 14 & 22 \end{bmatrix} \tag{7}$$

Notice the construction of the "strides" in Line 6 - the parameter of 2 told us to grab every second element from each row and each column.

# 2 Proofs

We will assume that no dataset contains repeated axes (i.e. no single tensor has two axes represented by the same graph), as this greatly affects the derived gradients. Shared axes - two tensors having one or more axes in common - are allowed. The case of shared axes is, after all, the whole point of developing this extension to prior work.

## 2.1 Permutations

Note that both $\text{vec}\left[\text{mat}_1\left[\mathcal{D}^\gamma\right]\right]$ and $\text{vec}\left[\text{mat}_\ell\left[\mathcal{D}^\gamma\right]\right]$ are row vectors containing the same elements, just in a different order. This means that there is a permutation matrix $\mathbf{P}_{\ell \to 1}$ such that $\text{vec}\left[\text{mat}_1\left[\mathcal{D}^\gamma\right]\right]^T \mathbf{P}_{\ell \to 1} = \text{vec}\left[\text{mat}_\ell\left[\mathcal{D}^\gamma\right]\right]^T$.

**Lemma 1** (Rearrangement lemma). $\mathbf{P}_{\ell \to 1}\left(\mathbf{I}_{d_{<\ell}} \otimes \mathbf{\Psi}_\ell \otimes \mathbf{I}_{d_{>\ell}}\right)\mathbf{P}_{\ell \to 1}^T = \mathbf{\Psi}_\ell \otimes \mathbf{I}_{d_{\setminus \ell}}$

*Proof.* While $\text{vec}$, $\text{mat}_\ell$ and $\bigotimes$ are defined as operations on matrices, for the purposes of permutations we can consider them as operations on indices. We can express them as follows:

$$\text{vec} : (i_1, ..., i_K) \to \left( \sum_\ell i_\ell d_{<\ell} \right) \tag{8}$$

$$\text{mat}_\ell : (i_1, ..., i_K) \to \left( i_\ell, \sum_{\ell' < \ell} i_{\ell'} d_{<\ell'} + \sum_{\ell' > \ell} i_{\ell'} \frac{d_{<\ell'}}{d_\ell} \right) \tag{9}$$

$$\bigotimes : \left( (i_1^1, i_1^2), ..., (i_K^1, i_K^2) \right) \to \left( \sum_\ell i_\ell^1 d_{<\ell}, \sum_\ell i_\ell^2 d_{<\ell} \right) \tag{10}$$

We'll consider just the rows of $\bigotimes$, $\bigotimes_{rows}$ - although the same argument applies with columns:

$$\bigotimes_{rows} : \left( i_1^1, ..., i_K^1 \right) \to \left( \sum_\ell i_\ell^1 d_{<\ell} \right) \tag{11}$$

Finally, we'll introduce the permutation operation $\sigma_{\ell \to 1}$ that will change the order of our Kronecker product:

$$\sigma_{\ell \to 1} : \left( (i_1^1, i_1^2), ..., (i_K^1, i_K^2) \right) \to \left( ((i_\ell^1, i_\ell^2), (i_1^1, i_1^2), ..., (i_{\ell-1}^1, i_{\ell-1}^2), (i_{\ell+1}^1, i_{\ell+1}^2), ..., (i_K^1, i_K^2) \right) \tag{12}$$

And again without loss of generality we restrict ourself to $\sigma_{\ell \to 1}^{rows}$:

$$\sigma_{\ell \to 1}^{rows} : \left( i_1^1, ..., i_K^1 \right) \to \left( i_\ell^1, i_1^1, ..., i_{\ell-1}^1, i_{\ell+1}^1, ..., i_K^1 \right) \tag{13}$$

After a Kronecker product our indices are in the form $\sum_\ell i_\ell d_{<\ell}$, and if we were to reorder it with $\sigma_{\ell \to 1}$ they would be in the form $i_\ell + \sum_{\ell' < \ell} i_{\ell'} d_{<\ell'} d_\ell + \sum_{\ell' > \ell} i_{\ell'} d_{<\ell'}$. Likewise, if we had matricized it we would have $\left( i_\ell, \sum_{\ell' < \ell} i_{\ell'} d_{<\ell'} + \sum_{\ell' > \ell} i_{\ell'} \frac{d_{<\ell'}}{d_\ell} \right)$, which is vectorized to $i_\ell + \sum_{\ell' < \ell} i_{\ell'} d_{<\ell'} d_\ell + \sum_{\ell' > \ell} i_{\ell'} d_{<\ell'}$. These reorderings are the same, and hence the matrix that represents it is $\mathbf{P}_{\ell \to 1}$.

$\square$

## 2.2 Derivation of the probability density function

Recall that the Kronecker-sum-structured normal distribution for a single tensor is defined as follows:

$$\text{vec}\left[\mathcal{D}^\gamma\right] \sim \mathcal{N}\left(\mathbf{0}, \left(\bigoplus_{\ell \in \gamma} \mathbf{\Psi}_\ell\right)^{-1}\right) \iff \mathcal{D}^\gamma \sim \mathcal{N}_{KS}\left(\{\mathbf{\Psi}_\ell\}_{\ell \in \gamma}\right) \tag{14}$$

The log-likelihood for this distribution is given in [6] for the matrix case and [5] for the general tensor case. However, neither of these papers provide a derivation. As the full derivation will motivated the construction of lemmas useful for the proofs of Theorems 1 and 2, we will give it here. First, observe that the density function is that of a normal distribution.

$$p\left(\mathcal{D}^\gamma\right) = \frac{\sqrt{\left|\bigoplus_{\ell \in \gamma} \mathbf{\Psi}_\ell\right|}}{(2\pi)^{\frac{d_\gamma^\gamma}{2}}} e^{\frac{-1}{2}\text{vec}[\mathcal{D}^\gamma]^T\left(\bigoplus_\ell \mathbf{\Psi}_\ell\right)\text{vec}[\mathcal{D}^\gamma]} \tag{15}$$

**Lemma 2** (⊕-vec lemma). $\text{vec}\left[\mathcal{D}^\gamma\right]^T \left(\bigoplus_\ell \mathbf{\Psi}_\ell\right) \text{vec}\left[\mathcal{D}^\gamma\right] = \sum_\ell \text{tr}\left[\mathbf{S}_\ell^\gamma \mathbf{\Psi}_\ell\right]$

*Proof.* This proof relies on the following two properties of vec: $(\mathbf{A} \otimes \mathbf{B})\text{vec}\left[\mathbf{C}\right] = \text{vec}\left[\mathbf{B}\mathbf{C}^T\mathbf{A}^T\right]$ and $\text{tr}\left[\mathbf{A}^T\mathbf{B}\right] = \text{vec}\left[\mathbf{A}\right]^T \text{vec}\left[\mathbf{B}\right]$. The $\mathbf{C}$ term picks up a transpose due to our use of the rows-first vectorization; when using columns-first notation the right hand side becomes $\text{vec}\left[\mathbf{B}\mathbf{C}\mathbf{A}^T\right]$.

$$\text{vec}\left[\mathcal{D}^\gamma\right]\left(\bigoplus_\ell \mathbf{\Psi}_\ell\right)\text{vec}\left[\mathcal{D}^\gamma\right] = \sum_\ell \text{vec}\left[\mathcal{D}^\gamma\right]^T \left(\mathbf{I}_{d_{<\ell}} \otimes \mathbf{\Psi}_\ell \otimes \mathbf{I}_{d_{>\ell}}\right)\text{vec}\left[\mathcal{D}^\gamma\right] \quad \text{(Definition of } \bigoplus\text{)}$$

$$= \sum_\ell \text{vec}\left[\text{mat}_1\left[\mathcal{D}^\gamma\right]\right]^T \left(\mathbf{I}_{d_{<\ell}} \otimes \mathbf{\Psi}_\ell \otimes \mathbf{I}_{d_{>\ell}}\right)\text{vec}\left[\text{mat}_1\left[\mathcal{D}^\gamma\right]\right] \tag{16}$$

$$= \sum_\ell \text{vec}\left[\text{mat}_\ell\left[\mathcal{D}^\gamma\right]\right]^T \mathbf{P}_{\ell \to 1}^T \left(\mathbf{I}_{d_{<\ell}} \otimes \mathbf{\Psi}_\ell \otimes \mathbf{I}_{d_{>\ell}}\right)\mathbf{P}_{\ell \to 1}\text{vec}\left[\text{mat}_\ell\left[\mathcal{D}^\gamma\right]\right]$$

$$\tag{17}$$

$$= \sum_\ell \text{vec}\left[\text{mat}_\ell\left[\mathcal{D}^\gamma\right]\right]^T \left(\mathbf{\Psi}_\ell \otimes \mathbf{I}_{d_{\backslash\ell}}\right)\text{vec}\left[\text{mat}_\ell\left[\mathcal{D}^\gamma\right]\right]$$

$$\text{(Rearrangement Lemma)}$$

$$= \sum_\ell \text{vec}\left[\text{mat}_\ell\left[\mathcal{D}^\gamma\right]\right]^T \text{vec}\left[\text{mat}_\ell\left[\mathcal{D}^\gamma\right]\mathbf{\Psi}_\ell^T\right] \tag{18}$$

$$= \sum_\ell \text{tr}\left[\mathbf{S}_\ell^\gamma \mathbf{\Psi}_\ell\right] \tag{19}$$

□

With this lemma, the probability density function in the single-tensor case can be expressed in the form:

$$p\left(\mathcal{D}^\gamma\right) = \frac{\sqrt{\left|\bigoplus_{\ell \in \gamma} \mathbf{\Psi}_\ell\right|}}{(2\pi)^{\frac{d_\gamma^\gamma}{2}}} e^{\frac{-1}{2}\sum_\ell \text{tr}\left[\mathbf{S}_\ell^\gamma \mathbf{\Psi}_\ell\right]} \tag{20}$$

Leading to the probability density function for the multi-tensor case as:

$$p\left(\{\mathcal{D}^\gamma\}\right) = \prod_\gamma \frac{\sqrt{\left|\bigoplus_{\ell\in\gamma}\mathbf{\Psi}_\ell\right|}}{(2\pi)^{\frac{d_\forall^\gamma}{2}}} e^{\frac{-1}{2}\sum_\ell \operatorname{tr}\left[\mathbf{S}_\ell^\gamma\mathbf{\Psi}_\ell\right]} \tag{21}$$

$$= \frac{\prod_\gamma \sqrt{\left|\bigoplus_{\ell\in\gamma}\mathbf{\Psi}_\ell\right|}}{(2\pi)^{\frac{d_\forall}{2}}} e^{\frac{-1}{2}\sum_\gamma\sum_\ell \operatorname{tr}\left[\mathbf{S}_\ell^\gamma\mathbf{\Psi}_\ell\right]} \tag{22}$$

$$= \frac{\prod_\gamma \sqrt{\left|\bigoplus_{\ell\in\gamma}\mathbf{\Psi}_\ell\right|}}{(2\pi)^{\frac{d_\forall}{2}}} e^{\frac{-1}{2}\sum_\ell \operatorname{tr}\left[\mathbf{S}_\ell\mathbf{\Psi}_\ell\right]} \tag{23}$$

The negative log-likelihood is thus:

$$\operatorname{NLL}\left(\{\mathcal{D}^\gamma\}\right) = \frac{d_\forall}{2}\log\left(2\pi\right) + \frac{1}{2}\sum_\ell \operatorname{tr}\left[\mathbf{S}_\ell\mathbf{\Psi}_\ell\right] - \frac{1}{2}\sum_\gamma \log\left|\bigoplus_{\ell\in\gamma}\mathbf{\Psi}_\ell\right| \tag{24}$$

## 2.3 Gradient

The derivation of the gradient of the negative log-likelihood is essentially the same as the derivation given by Kalaitzis et al. [6] for the original Bi-Graphical Lasso. Our derivation is complicated by the fact that we are considering general tensors rather than matrices. We'll let $\operatorname{sym}$ be the symmetricizing operator that must be applied as we are taking the derivative with respect to a symmetric matrix: $\operatorname{sym}\left[\mathbf{M}\right] = \mathbf{K}\circ\mathbf{M}$, where $\mathbf{K}$ is a matrix with 1s on the diagonal and 2s everywhere else. We'll also define $\mathbf{J}^{ij}$ to be the matrix of zeros except for a 1 at position $(i,j)$.

$$\frac{d}{d\mathbf{\Psi}_\ell}\mathrm{NLL}\left(\{\mathcal{D}^\gamma\}\right) = \frac{1}{2}\mathrm{sym}\left[\mathbf{S}_\ell\right] - \frac{1}{2}\sum_\gamma \frac{d}{d\mathbf{\Psi}_\ell}\log\left|\bigoplus_{\ell'\in\gamma}\mathbf{\Psi}_{\ell'}\right| \tag{25}$$

$$= \frac{1}{2}\mathrm{sym}\left[\mathbf{S}_\ell\right] - \frac{1}{2}\sum_\gamma \mathrm{tr}\left[\left(\bigoplus_{\ell'\in\gamma}\mathbf{\Psi}_{\ell'}\right)^{-1}\frac{d}{d\psi_\ell^{ij}}\bigoplus_{\ell'\in\gamma}\mathbf{\Psi}_{\ell'}\right]_{ij} \tag{26}$$

$$= \frac{1}{2}\mathrm{sym}\left[\mathbf{S}_\ell\right] - \frac{1}{2}\sum_\gamma \mathrm{tr}\left[\left(\bigoplus_{\ell'\in\gamma}\mathbf{\Psi}_{\ell'}\right)^{-1}\left(\mathbf{I}_{d_{<\ell}}\otimes\frac{d}{d\psi_\ell^{ij}}\mathbf{\Psi}_\ell\otimes\mathbf{I}_{d_{>\ell}}\right)\right]_{ij} \tag{27}$$

$$= \frac{1}{2}\mathrm{sym}\left[\mathbf{S}_\ell\right] - \frac{1}{2}\sum_\gamma \mathrm{tr}\left[\left(\bigoplus_{\ell'\in\gamma}\mathbf{\Psi}_{\ell'}\right)^{-1}\left(\mathbf{I}_{d_{<\ell}}\otimes\left(\mathbf{J}^{ij}+\mathbf{J}^{ji}-\delta_{ij}\mathbf{J}^{ij}\right)\otimes\mathbf{I}_{d_{>\ell}}\right)\right]_{ij} \tag{28}$$

$$= \frac{1}{2}\mathrm{sym}\left[\mathbf{S}_\ell\right] - \frac{1}{2}\sum_\gamma \left[(2-\delta_{ij})\mathrm{tr}\left[\left(\bigoplus_{\ell'\in\gamma}\mathbf{\Psi}_{\ell'}\right)^{-1}\left(\mathbf{I}_{d_{<\ell}}\otimes\mathbf{J}^{ij}\otimes\mathbf{I}_{d_{>\ell}}\right)\right]\right]_{ij} \tag{29}$$

$$= \frac{1}{2}\mathrm{sym}\left[\mathbf{S}_\ell\right] - \frac{1}{2}\sum_\gamma (2\mathbf{J}-\mathbf{I})\circ\mathrm{tr}\left[\left(\bigoplus_{\ell'\in\gamma}\mathbf{\Psi}_{\ell'}\right)^{-1}\left(\mathbf{I}_{d_{<\ell}}\otimes\mathbf{J}^{ij}\otimes\mathbf{I}_{d_{>\ell}}\right)\right]_{ij} \tag{30}$$

$$= \frac{1}{2}\mathrm{sym}\left[\mathbf{S}_\ell\right] - \frac{1}{2}\sum_\gamma \mathrm{sym}\left[\mathrm{tr}_{d_{>\ell}}^{d_{<\ell}}\left[\left(\bigoplus_{\ell'\in\gamma}\mathbf{\Psi}_{\ell'}\right)^{-1}\right]\right] \tag{31}$$

The MLE occurs when this gradient is zero, i.e. when the following equation is satisfied:

$$\mathbf{S}_\ell = \sum_\gamma \mathrm{tr}_{d_{>\ell}}^{d_{<\ell}}\left[\left(\bigoplus_{\ell'\in\gamma}\mathbf{\Psi}_\ell\right)^{-1}\right] \tag{32}$$

In other words, our effective Gram matrices are the best Kronecker-sum decomposition of the covariance matrix of the maximum likelihood estimate. Unfortunately, Kronecker-sum decomposition does not interact well with matrix inverses, so this does not directly yield an analytic solution. It does, however, yield a solution for the eigenvectors.

### 2.4 Maximum Likelihood Estimate for the Eigenvectors

We first produce two lemmas to aid in the derivation.

**Lemma 3** (Cyclic property of the stridewise-blockwise trace). *For any matrices* $\mathbf{M}$, $\mathbf{A}_{a\times a}$, $\mathbf{B}_{b\times b}$, *we have that* $\mathbf{tr}_b^a\left[\left(\mathbf{A}\otimes\mathbf{I}\otimes\mathbf{B}\right)\mathbf{M}\right] = \mathbf{tr}_b^a\left[\mathbf{M}\left(\mathbf{A}\otimes\mathbf{I}\otimes\mathbf{B}\right)\right]$

*Proof.* This follows directly from the cyclic property of the (normal) trace operator and the definition of the stridewise-blockwise trace. □

**Lemma 4** (Conjugacy extraction of the stridewise-blockwise trace). *For any matrices* $\mathbf{M}$ *and* $\mathbf{V}$, *we have that* $\mathrm{tr}_b^a\left[\left(\mathbf{I}_a\otimes\mathbf{V}\otimes\mathbf{I}_b\right)\mathbf{M}\left(\mathbf{I}_a\otimes\mathbf{V}\otimes\mathbf{I}_b\right)^T\right] = \mathbf{V}\mathrm{tr}_b^a\left[\mathbf{M}\right]\mathbf{V}^T$.

*Proof.*

$$\mathrm{tr}_b^a \left[ \left( \mathbf{I}_a \otimes \mathbf{V} \otimes \mathbf{I}_b \right) \mathbf{M} \left( \mathbf{I}_a \otimes \mathbf{V} \otimes \mathbf{I}_b \right)^T \right] = \left[ \mathrm{tr} \left[ \left( \mathbf{I}_a \otimes \mathbf{V} \otimes \mathbf{I}_b \right) \mathbf{M} \left( \mathbf{I}_a \otimes \mathbf{V} \otimes \mathbf{I}_b \right)^T \left( \mathbf{I}_a \otimes \mathbf{J}^{ij} \otimes \mathbf{I}_b \right) \right] \right]_{ij}$$

$$\text{(Definition of } \mathrm{tr}_b^a)$$

Thanks to the Rearrangement Lemma, we can get this just in terms of the standard blockwise trace, for which there exists a convenient lemma from Dahl et al. [3] that does the heavy lifting for us. Unfortunately, this requires inserting permutation matrices into every nook and cranny.

$$= \left[ \mathrm{tr} \left[ \mathbf{P} \left( \mathbf{I}_a \otimes \mathbf{V} \otimes \mathbf{I}_b \right) \mathbf{P}^T \mathbf{P} \mathbf{M} \mathbf{P}^T \mathbf{P} \left( \mathbf{I}_a \otimes \mathbf{V} \otimes \mathbf{I}_b \right)^T \mathbf{P}^T \mathbf{P} \left( \mathbf{I}_a \otimes \mathbf{J}^{ij} \otimes \mathbf{I}_b \right) \mathbf{P}^T \right] \right]_{ij} \quad (33)$$

$$= \left[ \mathrm{tr} \left[ \left( \mathbf{V} \otimes \mathbf{I}_{ab} \right)^T \mathbf{P} \mathbf{M} \mathbf{P}^T \left( \mathbf{V} \otimes \mathbf{I}_{ab} \right) \left( \mathbf{J}^{ij} \otimes \mathbf{I}_{ab} \right) \right] \right]_{ij} \quad (34)$$

$$= \mathrm{tr}_{ab} \left[ \left( \mathbf{V} \otimes \mathbf{I}_{ab} \right) \mathbf{P} \mathbf{M} \mathbf{P}^T \left( \mathbf{V} \otimes \mathbf{I}_{ab} \right)^T \right] \quad \text{(Definition of } \mathrm{tr}_{ab})$$

$$= \mathbf{V} \mathrm{tr}_{ab} \left[ \mathbf{P} \mathbf{M} \mathbf{P}^T \right] \mathbf{V}^T \quad \text{(Lemma 2 of Dahl et al. [3])}$$

We then can see analogously that $\mathrm{tr}_{ab} \left[ \mathbf{P} \mathbf{M} \mathbf{P}^T \right] = \mathrm{tr}_b^a \left[ \mathbf{M} \right]$, completing the proof.

$\square$

**Theorem 1.** *Let $\mathbf{V}_\ell \mathbf{e}_\ell \mathbf{V}_\ell^T$ be the eigendecomposition of $\mathbf{S}_\ell$. Then $\mathbf{V}_\ell$ are the eigenvectors of the maximum likelihood estimate of $\mathbf{\Psi}_\ell$.*

*Proof.*

$$\mathbf{S}_\ell = \sum_\gamma \mathrm{tr}_{d_{>\ell}}^{d_{<\ell}} \left[ \left( \bigoplus_{\ell' \in \gamma} \mathbf{\Psi}_\ell \right)^{-1} \right] \quad (35)$$

$$= \sum_\gamma \mathrm{tr}_{d_{>\ell}}^{d_{<\ell}} \left[ \left( \bigoplus_{\ell' \in \gamma} \mathbf{V}_\ell \mathbf{\Lambda}_\ell \mathbf{V}_\ell^T \right)^{-1} \right] \quad (36)$$

$$= \sum_\gamma \mathrm{tr}_{d_{>\ell}}^{d_{<\ell}} \left[ \left( \bigotimes_{\ell'} \mathbf{V}_{\ell'} \right) \left( \bigoplus_{\ell' \in \gamma} \mathbf{\Lambda}_\ell \right)^{-1} \left( \bigotimes_{ell'} \mathbf{V}_{\ell'} \right)^T \right] \quad (37)$$

$$= \sum_\gamma \mathrm{tr}_{d_{>\ell}}^{d_{<\ell}} \left[ \left( \mathbf{I}_{d_{<\ell}} \otimes \mathbf{V}_\ell \otimes \mathbf{I}_{d_{>\ell}} \right) \left( \bigoplus_{\ell' \in \gamma} \mathbf{\Lambda}_\ell \right)^{-1} \left( \mathbf{I}_{d_{<\ell}} \otimes \mathbf{V}_\ell \otimes \mathbf{I}_{d_{>\ell}} \right)^T \right] \quad \text{(Cyclic Property)}$$

$$= \sum_\gamma \mathbf{V} \mathrm{tr}_{d_{>\ell}}^{d_{<\ell}} \left[ \left( \bigoplus_{\ell' \in \gamma} \mathbf{\Lambda}_\ell \right)^{-1} \right] \mathbf{V}^T \quad \text{(Conjugacy Extraction)}$$

$$= \mathbf{V} \left( \sum_\gamma \mathrm{tr}_{d_{>\ell}}^{d_{<\ell}} \left[ \left( \bigoplus_{\ell' \in \gamma} \mathbf{\Lambda}_\ell \right)^{-1} \right] \right) \mathbf{V}^T \quad (38)$$

We conclude the proof by observing that the central matrix is diagonal, and thus the right hand side constitutes an eigendecomposition of $\mathbf{S}_\ell$. Thus $\mathbf{S}_\ell$ and $\mathbf{\Psi}_\ell$ share eigenvectors. $\square$

## 2.5 Maximum Likelihood Estimate for the Eigenvalues

In the previous section, we derived the eigenvectors of the maximum likelihood estimate. While interesting (they correspond to the principal components of our data), we need the eigenvalues

to reconstruct $\boldsymbol{\Psi}_\ell$. Our strategy for this is to transform our data such that the precision matrices are diagonal, and estimate these diagonals. This transformation is stated in terms of the Tucker operator ($[\![\mathcal{D}^\gamma; \{\mathbf{V}_\ell^T\}_{\ell \in \gamma}]\!]$). In the case where $\mathcal{D}$ is a matrix, we have that $[\![\mathbf{D}]; \mathbf{V}_{rows}^T, \mathbf{V}_{cols}^T]\!] = \mathbf{V}_{rows}\mathbf{D}\mathbf{V}_{cols}^T$. While the definition of the Tucker operator can be given in terms of "n-mode products"[8], we will only use the following property relating the Tucker operator to matricizationKolda [7]:

$$
\begin{aligned}
\mathcal{Y} &= [\![\mathcal{X}]; \{\mathbf{M}_\ell\}\} \\
\implies \mathrm{mat}_\ell[\mathcal{Y}] &= \mathbf{M}_\ell \mathrm{mat}_\ell[\mathcal{X}] \left(\mathbf{M}_K \otimes ... \otimes \mathbf{M}_{\ell+1} \otimes \mathbf{M}_{\ell-1} \otimes ... \otimes \mathbf{M}_1\right)^T \qquad \text{(Kolda [7])}
\end{aligned}
$$

The Tucker operator is an important concept for our calculation of the eigenvalues, but it is only the existence of such an operator that is important for our work; we never need to calculate it.

**Lemma 5** (Eigendecompositions of the Kronecker-sum-structured normal distribution). *Suppose $\{\mathcal{D}^\gamma\} \sim \mathcal{N}_{KS}\left(\{\boldsymbol{\Psi}_\ell\}\right)$. Then $\left\{[\![\mathcal{D}^\gamma; \{\mathbf{V}_\ell^T\}]\!]\right\} \sim \mathcal{N}_{KS}\left(\{\boldsymbol{\Lambda}_\ell\}\right)$ and the effective Gram matrices of this distribution are given by the eigenvalues $\mathbf{e}_\ell$ of the effective Gram matrices $\mathbf{S}_\ell$ of the original distribution.*

*Proof.* We will prove this by showing that the probability density function is that of a Kronecker-sum-structured normal distribution with the given parameters.

In the first part of the proof, we will massage the probability density function into a convenient form - this does not depend on the Tucker decomposition, and holds for our original dataset as well.

$$
p([\![\mathcal{D}^\gamma; \{\mathbf{V}_\ell^T\}_{\ell \in \gamma}]\!]) = p(\{\mathcal{D}^\gamma\}) \tag{39}
$$

$$
= \frac{\prod_\gamma \sqrt{\left|\bigoplus_{\ell \in \mathcal{D}^\gamma} \boldsymbol{\Psi}_\ell\right|}}{(2\pi)^{\frac{d_\forall}{2}}} e^{\frac{-1}{2} \sum_\ell \mathrm{tr}[\boldsymbol{\Psi}_\ell \mathbf{S}_\ell]} \tag{40}
$$

$$
= \frac{\prod_\gamma \sqrt{\left|\bigoplus_{\ell \in \mathcal{D}^\gamma} \boldsymbol{\Lambda}_\ell\right|}}{(2\pi)^{\frac{d_\forall}{2}}} e^{\frac{-1}{2} \sum_\ell \mathrm{tr}[\mathbf{V}_\ell \boldsymbol{\Lambda}_\ell \mathbf{V}_\ell^T \mathbf{S}_\ell]} \tag{41}
$$

$$
= \frac{\prod_\gamma \sqrt{\left|\bigoplus_{\ell \in \mathcal{D}^\gamma} \boldsymbol{\Lambda}_\ell\right|}}{(2\pi)^{\frac{d_\forall}{2}}} e^{\frac{-1}{2} \sum_\ell \mathrm{tr}[\boldsymbol{\Lambda}_\ell \mathbf{V}_\ell^T \mathbf{S}_\ell \mathbf{V}_\ell]} \tag{42}
$$

$$
= \frac{\prod_\gamma \sqrt{\left|\bigoplus_{\ell \in \mathcal{D}^\gamma} \boldsymbol{\Lambda}_\ell\right|}}{(2\pi)^{\frac{d_\forall}{2}}} e^{\frac{-1}{2} \sum_\ell \mathrm{tr}[\boldsymbol{\Lambda}_\ell \mathbf{e}_\ell]} \tag{43}
$$

To complete the proof, we must show that $\mathbf{e}_\ell$ are the effective Gram matrices for $[\![\mathcal{D}_j; \{\mathbf{V}_\ell^T\}_{\ell \in \mathcal{D}_j}]\!]$. For brevity, let $\mathbf{V}_{\setminus\ell} = (\mathbf{V}_K \otimes ... \otimes \mathbf{V}_{\ell+1} \otimes \mathbf{V}_{\ell-1} \otimes ... \otimes \mathbf{V}_1)$.

$$
\mathbf{e}_\ell = \mathbf{V}_\ell^T \mathbf{S}_\ell \mathbf{V}_\ell \tag{44}
$$

$$
= \sum_{\ell' \in \gamma} \frac{1}{n} \sum_i^n \mathbf{V}_\ell^T \mathrm{mat}_\ell[\mathcal{D}_i^\gamma] \mathrm{mat}_\ell[\mathcal{D}_i^\gamma]^T \mathbf{V}_\ell \qquad \text{(Definition of } \mathbf{S}_\ell\text{)}
$$

$$
= \sum_{\ell' \in \gamma} \frac{1}{n} \sum_i^n \mathbf{V}_\ell^T \mathrm{mat}_\ell[\mathcal{D}_i^\gamma] \mathbf{V}_{\setminus\ell}^T \mathbf{V}_{\setminus\ell} \mathrm{mat}_\ell[\mathcal{D}_i^\gamma]^T \mathbf{V}_\ell \tag{45}
$$

$$
= \sum_{\ell' \in \gamma} \frac{1}{n} \sum_i^n \mathrm{mat}_\ell\left[[\![\mathcal{D}_j; \{\mathbf{V}_\ell^T\}_{\ell \in \mathcal{D}_j}]\!]\right] \mathrm{mat}_\ell\left[[\![\mathcal{D}_j; \{\mathbf{V}_\ell^T\}_{\ell \in \mathcal{D}_j}]\!]\right]^T \tag{46}
$$

This completes the proof.

□

Since this transformed data is still normally distributed with Kronecker-sum structure, we can use the previously derived gradient (Line 32):

$$\frac{d}{d\mathbf{\Lambda}_\ell}\mathrm{NLL}\left(\{\mathcal{D}^\gamma\}\right) = \mathbf{e}_\ell - \sum_\gamma \mathrm{tr}_{d>\ell}^{d<\ell}\left[\left(\bigoplus_{\ell'\in\gamma}\mathbf{\Lambda}_\ell\right)^{-1}\right] \tag{47}$$

This yields Theorem 2:

**Theorem 2.** *Let* $\{\mathbf{G}_\ell^\gamma\}$ *be matrices such that the expression* $\bigoplus_{\ell\in\gamma}\mathbf{G}_\ell^\gamma$ *is the best Frobenius-norm approximation of* $\left(\bigoplus_{\ell\in\gamma}\mathbf{\Lambda}_\ell^t\right)^{-1}$. *Then, for a learning rate* $\mu_t$, *gradient descent can be performed with the update equation* $\mathbf{\Lambda}_\ell^{t+1} = \mathbf{\Lambda}_\ell^t - \mu_t\left[\mathbf{e}_\ell - \sum_{\gamma|\ell\in\gamma}\mathbf{G}_\ell^\gamma\right]$. *As* $\mathbf{\Psi}_\ell$ *is positive definite,* $\mu_t$ *must be chosen to prevent* $\mathbf{\Lambda}_\ell^t$ *from becoming negative.*

This is convenient because we have reduced our optimization task from one with $\sum_\ell d_\ell^2$ parameters to one with $\sum_\ell d_\ell$ parameters.

# 3   Dependences

All tests and figures were generated on a Linux (Ubuntu 20.04) with an Intel Core i7 chip and 8GB of RAM. Along with our code, we provide an environment file (environment.yml) that contains full details of all the dependencies used. In our GitHub repository (https://github.com/NeurIPS-GmGM-Paper/GmGM), we give precise and simple instructions on how to create a conda environment with the same setup as ours. Most of the packages used were specific to the experiments we ran. The dependencies necessary for our algorithm were Python 3.9 and NumPy 1.23.5.

# 4   Experiments

## 4.1   Synthetic data

We generated random graphs by modelling each edge's probability of existing as being drawn from independent Bernoulli distributions. When estimating the runtimes, we ran all models five times and averaged out the results. When creating precision-recall curves, we averaged the results of fifty runs of the models. Due to space reasons, we omitted the precision-recall curves for the tensor-variate case in our main paper, so we provide this here in Figure 1.

## 4.2   COIL video

We downloaded the processed COIL-20[10] dataset, and tested our model on it. We wanted to see if our model could understand the structure of a video, which we expected to consist of two linear graphs (for the rows and columns, i.e. each row is connected only to its neighbor rows) and a circular graph (for the frames, because the video is of a 360° rotation). To generate these graphs, we ran our algorithm on the duck video from the dataset, and then greedily kept the largest edge from each vertex such that vertices in the final graph had at most two edges. If we shuffled our data (shuffle rows, columns, and frames) and try to reconstruct it with these graphs, we get mixed results (Figure 2).

We can put a numeric value to the reconstruction, by measuring the percentage of the time that our reconstructed edges connect two adjacent rows/columns/frames. We get an accuracy of 80% for the rows, 91% for the columns, and 99% for the frames. This hints that it is quite good at reconstructing frames of videos, but rows and columns are a harder task. This could be due to the specific characteristics of this video, in which there are a lot of rows that spend most of their time being mostly blue.

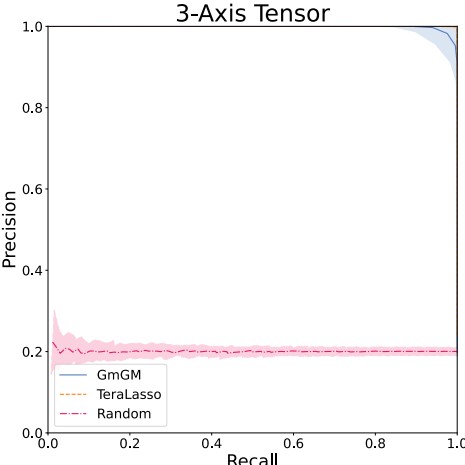

Figure 1: PR curves for the graphs generated from a 3-axis tensor. TeraLasso does almost perfectly; it can be hard to see as it is hugging the top right corner.

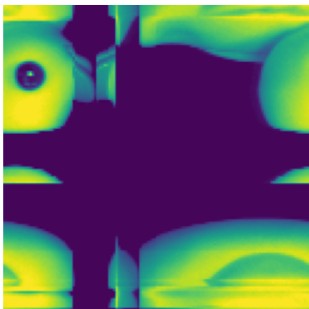

Figure 2: A reconstruction of the COIL-20 duck video after shuffling the rows, columns, and frames, using GmGM. While portions of the duck are well-reconstructed, it is clearly imperfect. Notably, the duck kisses itself.

### 4.3 EchoNet-Dynamic ECGs

We downloaded all of the EchoNet-Dynamic[11] data. This dataset did not have heartbeats labeled, so we picked a few videos at random and labeled them ourselves as a proof of concept. Specifically, we labeled every frame in which the mitral valve opened. Our goal was to see if the graphs produced by our algorithm could predict this opening. Table 1 contains the videos we picked, the labels we gave, and the labels we predicted.

Mitral valve predictions were done by taking GmGM's output frames graph in precision matrix form, and measuring the mass along the diagonals. We treated this as a time series (since each diagonal corresponds to an increasing time offset from all frames simultaneously). We applied gaussian blur and then a continuous wavelet transform peak detection algorithm[4] to find which diagonals had the most mass (Figure 3). These represent the offsets corresponding with a heartbeat. Given the first mitral valve opening and these offsets, we predict the remaining openings.

### 4.4 Mouse embryo stem cell transcriptomics

We used the mouse embryo stem cell dataset E-MTAB-2805[2]. This dataset had already been labeled by what stage of the cell cycle each cell was in. The data was log-transformed, and we restricted the

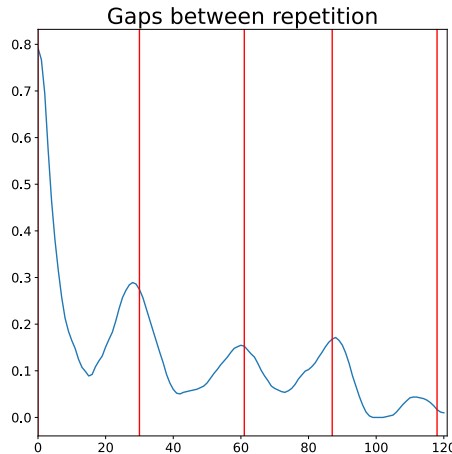

Figure 3: An example heartbeat offset detection, from EchoNet-Dynamic video 0XFE6E32991136338. The blue curve represents our Gaussian-blurred diagonal mass (if x=10, it represents the blurred mass of the 10th diagonal to the right of the main diagonal). The red lines represent the predicted peaks via a continuous wavelet transform peak detection algorithm. These represent offsets from the first mitral valve opening. For this video, the mitral valve opened on frame 17 and our first offset was on the 30th diagonal. Hence, we would predict the second mitral valve opening to occur at frame 47 (which, in this case, was correct).

gene set to the same mitosis-related genes used for Li et al. [9]'s analysis of this same dataset. We kept the top 100 edges in our output graphs for each vertex, and set the rest to zero.

### 4.5   10x Genomics flash frozen lymph node

For this experiment, we looked at a single-cell RNA-sequencing+ATAC-sequencing dataset from 10x Genomics[1]. We wanted to know whether clusters in UMAP-space make sense when viewed on GmGM's predicted graphs, whether clusters on the graphs made sense in UMAP-space, and whether these clusters had any meaning. Before performing the experiment, we removed cells whose library size was three median absolute deviations from the median, and similarly removed genes and peaks if the the total amount of times they were expressed was three median absolute deviations from the median. In our output graphs, we kept the top 5 edges per vertex.

From Figures 4 and 5, we can see that the clusters indeed seem to make sense in both UMAP-space and on the GmGM graph, as they all form coherent regions in both spaces.

To validate that these clusters are meaningful, we performed a GO term enrichment analysis; the full results of this analysis are saved on our GitHub repository, but we summarize them here.

Clusters 3 and 7 are clearly distinct in both spaces, and this is reflected in their GO terms. Cluster 3 was strongly associated with the CCKR signalling map and apoptosis, which none of the other clusters were. Cluster 7 was the most distinct, associated with the integrin signalling pathway, blood coagulation, and insulin. The other clusters all related to B and T cell-specific pathways. GmGM always grouped clusters 4 and 6 together, whereas UMAP would sometimes prefer to group cluster 6 with the rest of the clusters (compare Figures 5a and 6).

### 4.6   LifeLines-DEEP metagenomics + metabolomics

We used the LifeLines-DEEP metagenomics and metabolomics datasets[13]. We did not do any pre-processing to the metabolomics, and we used the already pre-processed version of the metagenomics data from Prost, Gazut, and Brüls [12]. We kept only patients that appeared in both datasets, and

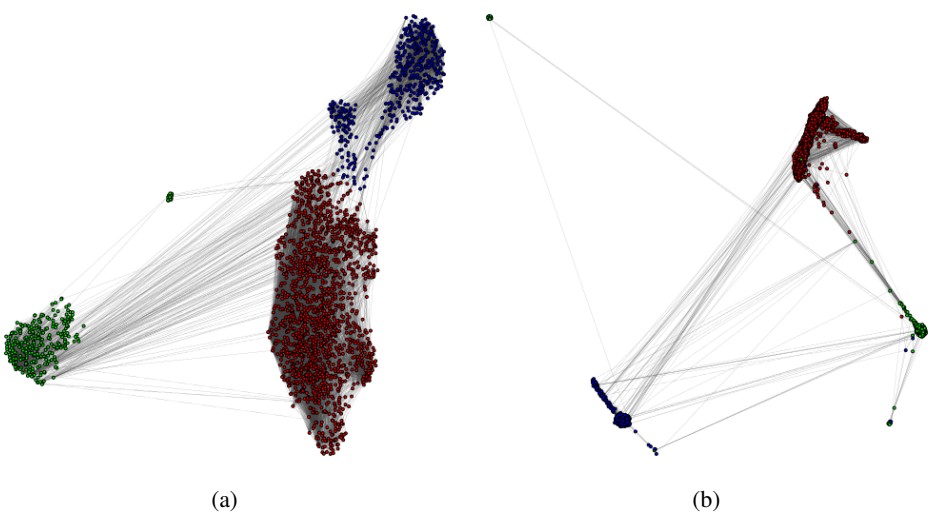

(a)                                                          (b)

Figure 4: (a) UMAP of the cells in the 10x Genomics dataset. Colored by kmeans (k=3). (b) GmGM's predicted graph over those cells, colored using the same clusters as on UMAP and plotted using igraph without reference to the outputs of UMAP.

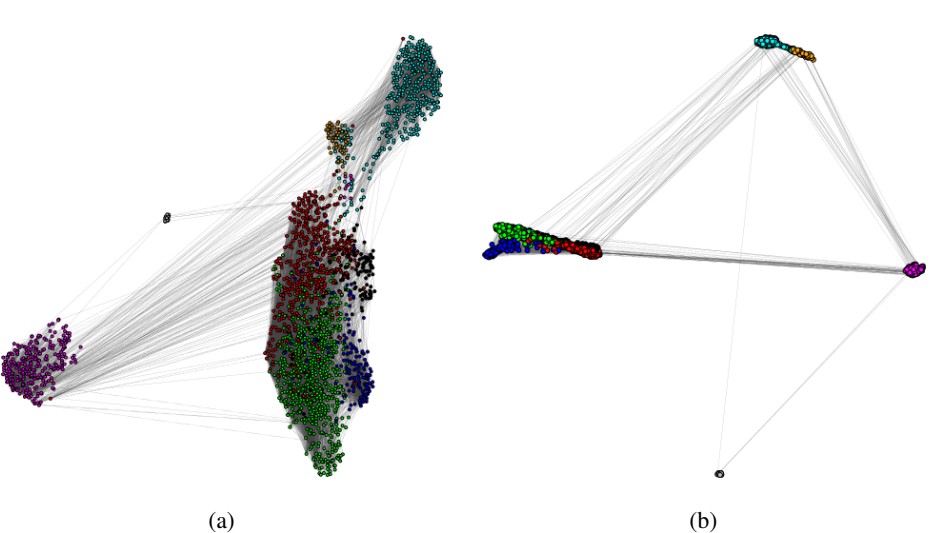

(a)                                                          (b)

Figure 5: (a) UMAP of the cells in the 10x Genomics dataset. Colored using same clusters as GmGM. (b) GmGM's predicted graph over those cells, colored using Louvain clustering.

| Video ID | Label | Predicted | Precision Matrix |
|---|---|---|---|
| | | | Echocardiogram Frames  |
| 0XFE6E32991136338 | [17, 47, 77, 106] | [17, 47, 78, 104] | Echocardiogram Frames  |
| 0XF072F7A9791B060 | [24, 56, 100] | [24, 59, 90] | Echocardiogram Frames  |
| 0XF70A3F712E03D87 | [22, 66, 110] | [22, 67, 111] | Echocardiogram Frames  |
| 0XF60BBEC9C303C98 | [19, 67, 114, 162] | [19, 66, 115, 162] | Echocardiogram Frames  |
| 0XF46CF63A2A1FA90 | [25, 79, 134, 188] | [25, 80, 133, 184] | |

Table 1: Mitral valve labellings and precision matrices for the EchoNet-Dynamic dataset. The precision matrices, for the most part, seem to have clear off-diagonal structures, as expected, and the mitral valve prediction is generally quite good; it is only significantly off for the last opening in 0XF072F7A9791B060.

log-transformed the data. We compared our model's results to the model given by Prost, Gazut, and Brüls [12] in the main paper.

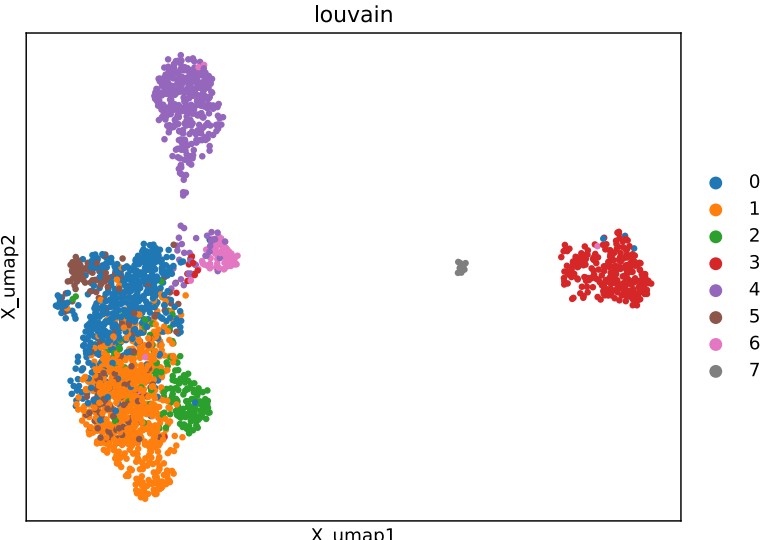

Figure 6: Another UMAP plot showing the same concept as Figure 5a, with the clusters labeled

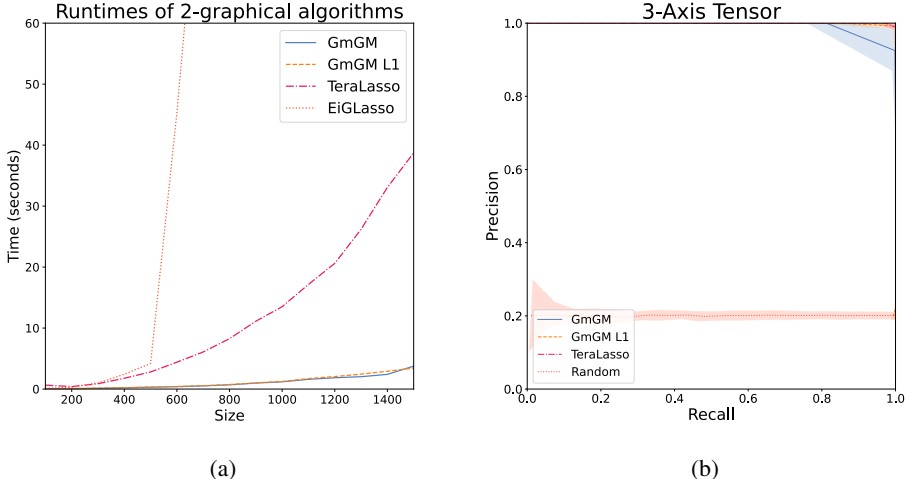

(a)                                    (b)

Figure 7: (a) Runtimes of our algorithm and prior work on matrix-variate data. Our regularized algorithm is denoted "GmGM L1", and takes about the same time as the unregularized "GmGM". (b) Precision-recall curves for tensor-variate data. TeraLasso and our regularized "GmGM L1" perform almost perfectly.

## 5   Regularization

As remarked in the main paper, our algorithm by default includes no regularization. This is because our algorithm leverages the fact that we have a closed-form expression for the eigenvectors of the maximum likelihood estimate to avoid costly eigendecompositions every iteration. We do not have a closed-form expression for the eigenvectors in the regularized case.

Nevertheless, we can add regularization to the eigenvalue estimation by performing an eigenrecomposition and regularizing that. Eigenrecomposition requires a matrix multiplication, which is quite costly compared to the cost of an unregularized iteration - both in practice, and asymptotically in the matrix-variate case (matrix multiplication is $O(\sum_\ell d_\ell^3)$ whereas an unregularized iteration is

$O(\prod_\ell d_\ell)$). Thus, to regularize we first let our algorithm converge to the MLE before considering the penalty term. This allows us to avoid a major increase in runtime; our regularized algorithm runs in roughly the same time as the unregularized one (Figure 7a).

It is important to note that this estimator is slightly different than the standard Lasso estimator, as the standard estimator would minimize $\|\boldsymbol{\Psi}_\ell\|_1$ and our estimator minimizes $\|\hat{\mathbf{V}}_\ell \boldsymbol{\Lambda}_\ell \hat{\mathbf{V}}_\ell^T\|_1$ (where only the eigenvalues $\boldsymbol{\Lambda}_\ell$ are free to vary). It can be derived as follows:

$$\frac{\partial}{\partial \lambda_i}\|\mathbf{V}\boldsymbol{\Lambda}\mathbf{V}^T\|_1 = \frac{\partial}{\partial \lambda_i}\|\sum_j \lambda_j v_{ja} v_{bj}\|_1 \tag{48}$$

$$= \left[\frac{\partial}{\partial \lambda_i}\left|\sum_j \lambda_j v_{ja} v_{bj}\right|\right]_{ab} \tag{49}$$

$$= \left[\frac{\partial}{\partial \lambda_i}\mathrm{sign}\left[\sum_j \lambda_j v_{ja} v_{bj}\right] v_{ia} v_{bi}\right]_{ab} \tag{50}$$

$$= \left[\mathrm{sign}\left[\mathbf{V}\boldsymbol{\Lambda}\mathbf{V}^T\right]_{ab} v_{ia} v_{bi}\right]_{ab} \tag{51}$$

$$= \mathbf{v}_i^T \mathrm{sign}\left[\mathbf{V}\boldsymbol{\Lambda}\mathbf{V}^T\right] \mathbf{v}_i \tag{52}$$

Despite this difference, it performs comparably to prior work. We show in Figure 7b the precision-recall curves for the 3-axis case, and observe that it performs almost perfectly. This is notable as it was the case that the unregularized algorithm performed worse than prior work.