# OpenReview forum: "GmGM: a fast Gaussian graphical model for multi-modal data"
_NeurIPS.cc/2023/Conference — Submitted to NeurIPS 2023_

### Official Review · Reviewer_ooWU · 2023-06-16

**Soundness:** 3 good
**Presentation:** 3 good
**Contribution:** 2 fair
**Rating:** 6
**Confidence:** 4

**Summary:**

This paper introduces the Gaussian multi-Graphical Model (GmGM) as a novel method to construct sparse graph representations of matrix- and tensor-variate data. It simultaneously learns the representation across several tensors that share axes. The authors demonstrate that GmGM outperforms previous methods in terms of speed when applied to matrix data.

**Strengths:**

1.	GmGM extends the application of Gaussian Graphical Models to multi-tensor datasets, presenting a novel approach in the field.

2.	GmGM exhibits significantly improved speed compared to previous methods when dealing with matrix data.

3.	The results of GmGM on five real datasets are well explained.

4.	Especially, I appreciate the comprehensive discussion provided in this paper. The authors present cases where the results are excellent, as well as cases where the results are not as impressive, such as the performance on higher-order tensor data (fig 4b) and the E-MTAB-2805 dataset (fig 6a). This in-depth analysis helps readers gain a better understanding of the method and be aware of the situations in which it should be employed.

**Weaknesses:**

One major concern I have relates to the evaluation. Although the authors present many intriguing findings on the datasets, it would be beneficial to include some more quantitative analysis.

**Questions:**

1.	Could the authors show the results on the COIL-20 dataset and provide quantitative comparisons with baselines in terms of both efficiency and accuracy?
2.	The two multi-omics datasets, LifeLines-DEEP and 10x, are analyzed from different perspectives. It would be advantageous if the authors could also present UMAP consistency analysis results for the LifeLines-DEEP dataset. Additionally, conducting quantitative comparisons with baselines on the 10x dataset would also be informative for readers.


**Limitations:**

Yes. It is well discussed in the study.

---

> ### Author Rebuttal · Authors · 2023-08-08
>
> Thank you for your comments!
>
> -	“Could the authors show the results on the COIL-20 dataset and provide quantitative comparisons with baselines in terms of both efficiency and accuracy?”
>
> We are very happy to provide quantitative results to all experiments, which we have run on the same computer as in our paper (Ubuntu 20.04 with Intel Core i7 Processor and 8GB RAM).
>
> COIL-20 Duck Performance in terms of row/col/frame recovery accuracies and runtime:
>
> GmGM: 80%/91%/99% in 0.14 seconds
>
> TeraLasso: 80%/91%/99% in 1.99 seconds
>
> We had not measured runtime for TeraLasso on this dataset before now.  On synthetic data, as reported in the paper, our model improved efficiency compared to TeraLasso in the 3-axis tensor case, but not this much.   This was a pleasant surprise!  We suspect this is due to real data tending to require more iterations to converge, in which case the computation of Gram matrices no longer dominates the runtime for 3-axis data.  Rather, the speed of iteration will dominate - and our algorithm has much faster iterations than TeraLasso due to it avoiding an eigendecomposition each time.  If you had not suggested this comparison, we would not have noticed this – thank you!
>
> -	“[…] conducting quantitative comparisons with baselines on the 10x dataset would also be informative for readers […]”
>
> Quantitative analysis on the 10x dataset is harder, as there are no ground truth (cell labels), preventing us from performing quantitative analysis such as measuring assortativity (as in the LifeLines-DEEP experiment), nor can we show the performance in terms of recovery accuracy.
>
> However, we can compare our algorithm's runtime with that of TeraLasso, as below.  We did not show the EiGLasso runtime as it was too slow.
>
> GmGM: 94.39 seconds
>
> TeraLasso: 3752.57 seconds
>
> For comparison, we created a UMAP consistency plot for both GmGM and TeraLasso, in the global response (Figure R1).
>
> -	“It would be advantageous if the authors could also present UMAP consistency analysis results for the LifeLines-DEEP dataset.”
>
> For this dataset, we kept the top 1200 largest-weighted edges of the estimated graph in accordance with the paper we used as a baseline ("A zero inflated log-normal model for inference of sparse microbial association networks" by Prost et al.), whose model is called 'ZiLN'.  Note that ZiLN can only learn the species graph; it makes an independence assumption for the genes.  Our model and TeraLasso make no independence assumption and simultaneously learn multiple graphs.
>
> We have given two examples of our algorithm, one in which we only consider metagenomics (as in ZiLN and TeraLasso), and one in which we consider multiple modalities (metagenomics and metabolomics) simultaneously.  ZiLN and TeraLasso are not able to consider multiple modalities; in this case, only GmGM could be run.
>
> ZiLN: 3.2 seconds (learns only species graph)
>
> GmGM: 2.59 seconds (learns species and people graphs)
>
> GmGM: 22.18 seconds (learns species, people, and metabolomics graphs)
>
> TeraLasso: 1299.33 seconds (learns species and people graphs)
>
> From a speed perspective, we can see that we greatly outperform prior multi-axis work (TeraLasso), and perform favorably to single-axis work, especially on a per-graph basis:
>
> ZiLN (species): 3.2 seconds per graph
>
> GmGM (species, people): 1.30 seconds per graph
>
> GmGM (species, people, metabolomics): 7.39 seconds per graph
>
> TeraLasso (species, people): 649.67 seconds per graph
>
>
> It is quite encouraging that we have managed to outperform a single-axis method in one scenario, given the fact that single-axis methods can take advantage of stronger assumptions to make simplifications.
>
> The UMAP consistency plot is given in the global response document (Figure R2); we can see that the clusters we find also happen to correspond to distinct regions in UMAP-space.

---

> > ### Comment · Reviewer_ooWU · 2023-08-15
> >
> > Thanks for these responses, which effectively address my concerns regarding the quantitative evaluation.

---

### Official Review · Reviewer_KacZ · 2023-06-23

**Soundness:** 2 fair
**Presentation:** 2 fair
**Contribution:** 2 fair
**Rating:** 5
**Confidence:** 3

**Summary:**

The authors propose Gaussian multi-Graphical Model (GmGM), a novel approach to constructing sparse graph representations of matrix- and tensor-variate data. It stands out from previous models by learning representations across multiple tensors that share axes simultaneously, a feature crucial for analyzing multimodal datasets, particularly in multi-omics scenarios. The GmGM algorithm utilizes a single eigendecomposition per axis, which results in a significant speedup over previous models. This efficiency enables the application of the methodology on large multi-modal datasets, such as single-cell multi-omics data, a task that was challenging with previous approaches.

**Strengths:**

1. Fair and Interesting Motivation:
The paper's motivation on model multi-tensor decomposition with shared axis is rooted in the real-world need for handling multi-omics scenarios, which often involve multi-tensor data with shared axes. The GmGM is introduced as a solution, addressing a significant gap in existing data analysis methodologies and providing a fair and interesting motivation for the study.

2. Reasonable solution and impressive improvements in Efficiency
The GmGM model stands out for its impressive efficiency improvements, achieved through the use of the KS decomposition of the precision matrix and transiting it to the eigen-decomposition over each dim. This approach results in a substantial speedup over previous models, enabling the handling of large multi-modal datasets, This efficiency, coupled with the model's ability to maintain state-of-the-art performance, underscores the strength of the paper.

**Weaknesses:**

1. **Limited Technical Contribution**

While the problem setting proposed in the paper is reasonable, the algorithm's strict assumptions about data integrity (no missing data) and quality (no noise) somewhat limit its potential for broader application. The authors are encouraged to consider relaxing these assumptions or proposing strategies to handle missing data and noise, which are common issues in real-world datasets. Addressing these issues could significantly enhance the model's practical utility and broaden its applicability.

2. **Improvements Needed in Representation and Flow**

The paper could benefit from substantial improvements in its representation and flow. The omission of important concepts and content significantly hinders reader comprehension. Some sentences appear casual and can lead to confusion. The overall logical flow of the paper is not clear, making it difficult to follow. This is particularly evident in the following areas:

   - Concepts such as the Kronecker product and Gram matrix are not clearly introduced.
   - Many notations and their subscripts and superscripts in the algorithm table are not clearly defined.
   - The task setting and metric definition in the experimental section are vague, reducing the persuasiveness of the validation part.

Overall, the authors are encouraged to make a concerted effort to reorganize and polish the paper's presentation, improve the flow, and highlight the key points of the work and problem. This could significantly enhance the readability and impact of the paper.

**Questions:**

See weakness parts

**Limitations:**

See weakness parts

---

> ### Author Rebuttal · Authors · 2023-08-08
>
> Thank you for taking the time to review our paper!
>
> -	“[…] strict assumptions about data integrity (no missing data) […] limit […] broader application […]”
>
> This is a very fair point.  To address the problem of missing data, it is helpful to split it into two cases:
>
> 1) Elements of one or more of the input tensors are missing
>
> As a strategy to handle missing data, we would propose doing complete cases analysis, or imputation.   Our aim in this paper is to take an already existing class of algorithms, and make it practically usable - before, none of these algorithms could be used on anything but the smallest datasets.  This is why we did not spend too much time addressing concerns about missing data, which were also not addressed by the aforementioned models.
>
> Embedding the imputation of missing data into our approach would be a very powerful addition to our software implementation.
>
> 2) Two tensors share an axis, but do not contain exactly the same elements
>
> This type of missing data has been addressed and discussed in our paper (see Section 4, “Limitations”).
>
>
> -	“[…] strict assumptions about […] quality (no noise) […] limit […] broader application […]”
>
>
> Finally, you mention the case in which additional noise is added.  This is an interesting case, which we had not considered before.  We didn’t perform robustness-against-noise tests on simulated data, but our performance on real data suggests that our methodology is effective even when not explicitly modelling the noise.  We will add an experiment to the paper in which we explore the addition of different levels of noise and its effect on graph recovery.
>
> This has not been considered in the literature we are exploring (multi-axis models).   In the single-axis case, the paper "A Nonconvex Variational Approach for Robust Graphical Lasso" by Benfenati et al. proposes handling this by including an additional regularization term and a small modification to the loss function.  At first glance, their approach seems amenable to the treatment given in this paper (specifically an analog of Theorem 1 may hold); but this would be left to future work.
>
>
> -	“Concepts such as the Kronecker product and Gram matrix are not clearly introduced […] Many notations and their subscripts and superscripts in the algorithm table are not clearly defined.”
>
>
> We were too trigger-happy with offloading information into the supplementary material, and our paper's flow and cohesion has suffered because of it.  In the final version, we will address this.  (Concretely, we will include a small section at the start introducing more of the notation, concepts, and technical terms - with examples, when sensible).
>
> -	“The task setting and metric definition in the experimental section are vague”
>
> Task setting was mainly linked to a task’s use in prior work for the sake of comparison and the availability of ground truth for performance evaluation.  Not all datasets were amenable to quantitative analysis, as ground truth graphs are often unknown.  In the paper, we will clarify our rationale behind each task we performed, and the metric chosen for that task, as explained in the following;
>
> COIL-20 Duck Video: A limited version of this analysis, done on a heavily down-sampled and flattened version of the video, was performed in the original BiGLasso paper as a proof-of-concept, without quantitative analysis.  We chose the metric of row/column/frame recovery percent as it seemed the most natural way to measure our algorithm’s capability to reconstruct the video.
>
> LifeLines-DEEP:  The ZiLN paper performed the same analysis with the same metric, which we have repeated in the paper.
>
> Mouse Embryo Stem Cells: The scBiGLasso paper also considered this dataset.  Validation in this case is more complicated due to the underlying biology not having an obvious interpretation.  Thus, we opted to explore how well our algorithm could separate the three cell stages, rather than repeating the analysis in the scBiGLasso paper.
>
> Heartbeat Videos: This was not considered in prior work.  We chose this with the hopes that it would prove to be a similar but more complex version of the COIL-20 analysis due to the periodic nature of a heartbeat.  We show that our algorithm can capture this periodic nature quantitatively through the prediction of future heartbeats.
>
> 10x Genomics: This dataset is the largest that we have considered – prior work was not able to run on such a large dataset in a reasonable amount of time, as shown in the following table.
>
> GmGM: 94.39 seconds
>
> TeraLasso: 3752.57 seconds
>
> As there was no ground truth available, we could not perform quantitative analysis on the 10x dataset, and thus we compared the similarity of clusters found on our graph to structure found using a well-known nonlinear transformation technique, UMAP.

---

> > ### Comment · Reviewer_KacZ · 2023-08-16
> >
> > I appreciate the authors' hard work on the response. It somehow addresses my concerns, but I still think the work could be further polished and may not change the score now.

---

> > > ### Author Response · Authors · 2023-08-16
> > >
> > > No worries!  Which areas of the paper would you recommend need the most polishing?

---

### Official Review · Reviewer_pfpR · 2023-07-01

**Soundness:** 2 fair
**Presentation:** 3 good
**Contribution:** 3 good
**Rating:** 6
**Confidence:** 3

**Summary:**

This paper proposes the Gaussian multi-Graphical Model, a novel method to extend the use of Gaussian Graphical Models to multi-tensor datasets. It generalizes Gaussian graphical models to the common scenario of multi-tensor datasets. For the single-tensor case,  the proposed algorithm is faster than prior work while still preserving state-of-the-art performance.

**Strengths:**

The paper considers an interesting and still challenging topic, extending conventional Gaussian graphical models (GGM) for complex systems like multi-modal data models. The paper has been generally well-written and the problem has been clearly defined. Indeed, the theoretical parts that extend the GGM to multi-tensor datasets have proper quality.  This algorithm is significantly faster on lower-order tensor data (reported for the synthetic data sets) and its efficacy is slightly better in the real-world data sets.

**Weaknesses:**

- Some parts of the paper should be checked again. For instance, line 52 starts to explain the computational costs of the state-of-the-art methods. The parameters n and p have not been defined before. It seems it uses the defined parameter in the main reference paper (Kalaitzis et. al. 2013), where n and p are the numbers of observations and features, respectively. Indeed, the computational costs of the other baselines need a piece of clarification. For instance, O(n^2 * p^2) in BIGLasso represents the number of non-zeros in the Kronecker-sum (KS) structure. It would be better if the authors consider the full cost of the algorithm for the proposed method and available baselines.

- The paper models each tensor as being drawn independently
 from a Kronecker-sum normal distribution. It makes sense to see this assumption reduces the computational cost at least in small-order data sets. However, it does not describe how this strong assumption still preserves state-of-the-art performance.

- As has been reported in the paper, the proposed solution can not improve the complexity of higher-order tensor data sets (fig. 4b). Indeed, its performance can not significantly outperform the other baseline (Fig. 5a). By decreasing the sparsity, the performance of the model suffers and it seems it works properly only on high sparse graphs (Fig 7).

**Questions:**

- See the Weaknesses part.

-  A question about the scope of the proposed model: It has been designed for multi-modal data sets. Another problem that has a similar structure is distributed learning when the entire data set is divided into several partition and each partition provides local inference. The partitions are clusters of multi-dimensional data sets, and the features are the same for all clusters. Can the proposed method be used for estimating the conditional dependencies between the features and also dependencies between local partitions?

**Limitations:**

The authors addressed the limitation of the work in the paper.

---

> ### Author Rebuttal · Authors · 2023-08-08
>
> Thank you for your review!
>
> -	“[…] line 52 […] The parameters n and p have not been defined before. […]”
>
> The reviewer is correct; we used the same notation as in Kalaitzis et al; in the final version, we will clarify this.  We will use the following notation instead: Let $d_i$ be the size of the ith axis (so a 50 by 60 matrix would have $d_1 = 50, d_2 = 60$).
>
> -	“$O(n^2p^2)$ in BIGLasso represents the number of non-zeros in the Kronecker-sum (KS) structure”
>
> In general, the number of non-zeros when using a Kronecker Sum structure is $O(d_1d_2(d_1 + d_2))$.  $O(d_1^2d_2^2)$ refers to the number of non-zeros if we were to use a Kronecker Product structure.
>
> -	“the computational costs […] of the algorithm for the proposed method and available baselines […]”
>
> For space complexity, all algorithms (ours and prior work) except BiGLasso achieve an optimal $O(\sum_i d_i^2)$.  BiGLasso does not report space complexity, but by looking at the implementation we know it must be at least $O(d_1^2d_2 + d_2^2d_1)$.
>
> In the following, we give the computational complexities for each algorithm we compared against.  To keep the notation simple, we assume all axes are the same size ($d = d_1 = d_2 = ... = d_K$), where $K$ is the number of axes.  Note that BiGLasso and EiGLasso only work on matrix data ($K=2$).
>
> •	BiGLasso: $O(Kd^4)$ per iteration (specifically, $O(Kd)$ Lasso regressions per iteration).
>
> •	EiGLasso: Does not explicitly state in the paper, but can be seen to be $O(Kd^3)$ per iteration due to the use of eigendecompositions.
>
> •	TeraLasso: Does not explicitly state in the paper, but can be seen to be $O(Kd^3 + d^K)$ per iteration due to the use of eigendecompositions and projecting data onto the space of Kronecker-sum-decomposable matrices.  Computation of Gram matrices at the start is $O(Kd^{K+1})$.
>
> •	GmGM (our work): $O(Kd^{K+1})$ overall due to computing the Gram matrices and eigendecompositions at the start.  Per iteration, however, it is $O(d^K)$ due to projecting data onto the space of Kronecker-sum-decomposable matrices.
>
>
> - “[…] it does not describe how this strong assumption still preserves state-of-the-art performance […]”
>
> For unimodal data, we are making the same assumption (Kronecker sum) as prior multi-axis work (BiGLasso, scBiGLasso, EiGLasso, TeraLasso), and a weaker assumption than prior single-axis work (Graphical Lasso, which assumes full independence).
>
> We are the first to consider multiple modalities for multi-axis data, so there is no prior work to compare against in this case.
>
> - “[…] the proposed solution can not improve the complexity of higher-order tensor data sets […]”
>
> The most common type of tensor data is 2-axis (matrix) data, in which we achieve a very substantial speedup (i.e. an order of magnitude) compared to prior work.  While there is not as dramatic a speedup for 3-axis data, we are still faster than prior work.  The main barrier is the Gram matrix computation, which is effectively a preprocessing step that all these algorithms have to do.
>
> Furthermore, despite our similar performances on synthetic data, we did find that our algorithm was much faster than prior work on real-world 3-axis data as well, as reported below.
>
> COIL-20 Duck Performance: (row/col/frame accuracies)
>
> GmGM: 80%/91%/99% in 0.14 seconds
>
> TeraLasso: 80%/91%/99% in 1.99 seconds
>
> We suspect this is due to real data requiring more iterations to converge, and hence the Gram matrix computation no longer dominates as it was during our synthetic data experiments.
>
> -	“the model […] works properly only on high sparse graphs (Fig 7)”
>
> Yes, this is true, sparsity is very important.  A sparse precision matrix defines a Gaussian Markov random field, which is conventionally represented by a weighted, undirected graph (“Graphical models”, Lauritzen, 1996) – our algorithm, and prior work, fits in this framework.  The assumptions we and prior work make (Kronecker sum distribution) and the type of graph we learn (conditional dependencies) both encourage sparsity, so we will perform better when this assumption is met.  This is true for all models considered in the paper.
>
> -	Questions (distributed learning)
>
> This is a very interesting question!  We have not tested the model in this situation, but from what you describe it seems that the model fits the scenario well.  The iterative component of the algorithm does require the eigenvalues of the Gram matrices from each partitions, but the rest of the algorithm can be performed locally wherever the partition is stored.  Thus it should still be applicable in situations in which data is being kept in separate centers for privacy/data protection reasons.
>
> The downside is that this will only find connections within each local partition, not between elements in different partitions.  Depending in the scenario this may or may not be acceptable.  The model would do a good job of finding connections between features still.  (The output would be one graph of connections between features, learned globally, and then for each partition one graph of connections between samples in that partition).
>
> As an example, suppose a hospital in Belgium and a hospital in Sri Lanka both collect the same healthcare information on patients.  It is unlikely for there to be connections between a patient in Belgium and a patient in Sri Lanka, so this model would be reasonable.  However, if instead the data was all collected in from one town Belgium, being partitioned after the fact, then the assumptions the model makes could be too strong, as there could plausibly be conditional dependencies between patients in different partitions.

---

> > ### Comment · Reviewer_pfpR · 2023-08-16
> >
> > Thanks for the author's response and for addressing the concerns
> > (one hint: graphical Lasso generally assumes a joint Gaussian distribution between features and not full independence. It estimates the inverse of the full covariance matrix).
> >
> > I keep my initial score because, despite the shortages in some parts, the paper still has proper potential.

---

> > > ### Author Response · Authors · 2023-08-16
> > >
> > > Thank you for the kind words.  We did wish to clear up one potential misunderstanding;
> > >
> > > Graphical Lasso does indeed assume a joint Gaussian for the features and estimates the precision matrix, as you say - but it also assumes independence for the samples.
> > >
> > > (To fact-check this claim, one can refer to Section 2.1 of "Model Selection Through Sparse Maximum Likelihood Estimation for Multivariate Gaussian or Binary Data" by Banarjee et al.  The original GLasso paper, "Sparse inverse covariance estimation with the graphical lasso" by Friedman et al, follows Banarjee et al in their choice of distribution)
> > >
> > > Unlike GLasso, multi-axis methods do not make this independence assumption for the samples - instead we replace it with the weaker assumption that the features and samples both have dependencies, which interact through the Kronecker sum.

---

### Author Rebuttal · Authors · 2023-08-08

This is the global rebuttal; individual reviewer rebuttals have been submitted separately in accordance to the directions given to the authors.  The global rebuttal comprises of an attached pdf containing only figures and captions.

---

### Decision · Program_Chairs · 2023-09-21

**Decision:**

Reject

**Comment:**

During the discussion between the AC and the reviewers, none of the reviewers expressed a strong willingness to accept this paper.

During the rebuttal, some of the criticisms from the reviewers were properly addressed by the authors, such as computational complexity and some few additional experiments.

Unfortunately, some other issues were not resolved, such as the limited technical contribution. As a reviewer mentions, handling missing data and noise could be one possible way to improve the current paper.

Thus, while there is some technical contribution in terms of a Kronecker-sum model and a simple algorithm using a single eigendecomposition of the data per axis, this works only for unregularized estimation as in Theorem 1 (or at most for regularization that plays with the data eigenvectors as discussed in Section 5 in the Appendix). As also acknowledged by the authors in Section 4 (Limitations), using more general regularizers (e.g., L1 norm for sparsity) would require one eigendecomposition per iteration, thus losing some advantage in computational complexity. Therefore, the AC finds there is not enough to grant acceptance of this paper.

As a final point, perhaps a more systematic/thorough quantitative experimental evaluation could strengthen the paper by testing the proposed algorithm (versus other algorithms) in the context of real-world multi-modal data, in terms of test log-likelihood.